# Evaluation of chromatin accessibility in prefrontal cortex of individuals with schizophrenia

Julien Bryois [1], Melanie E. Garrett[2], Lingyun Song[3], Alexias Safi[3], Paola Giusti-Rodriguez [4], Graham D. Johnson [3], Annie W. Shieh[13], Alfonso Buil[5], John F. Fullard[6], Panos Roussos [6,7,8], Pamela Sklar[6], Schahram Akbarian [6], Vahram Haroutunian [6,9], Craig A. Stockmeier [10], Gregory A. Wray[3,11], Kevin P. White[12], Chunyu Liu[13], Timothy E. Reddy [3,14], Allison Ashley-Koch[2,15], Patrick F. Sullivan [1,4,16] & Gregory E. Crawford [3,17]

Schizophrenia genome-wide association studies have identified >150 regions of the genome associated with disease risk, yet there is little evidence that coding mutations contribute to this disorder. To explore the mechanism of non-coding regulatory elements in schizophrenia, we performed ATAC-seq on adult prefrontal cortex brain samples from 135 individuals with schizophrenia and 137 controls, and identified 118,152 ATAC-seq peaks. These accessible chromatin regions in the brain are highly enriched for schizophrenia SNP heritability. Accessible chromatin regions that overlap evolutionarily conserved regions exhibit an even higher heritability enrichment, indicating that sequence conservation can further refine functional risk variants. We identify few differences in chromatin accessibility between cases and controls, in contrast to thousands of age-related differential accessible chromatin regions. Altogether, we characterize chromatin accessibility in the human prefrontal cortex, the effect of schizophrenia and age on chromatin accessibility, and provide evidence that our dataset will allow for fine mapping of risk variants.

[1] Department of Medical Epidemiology and Biostatistics, Karolinska Institutet, SE-17177 Stockholm, Sweden. [2] Duke Molecular Physiology Institute, Durham, NC 27701, USA. [3] Center for Genomic and Computational Biology, Duke University, Durham, NC 27708, USA. [4] Department of Genetics, University of North Carolina, Chapel Hill, NC 27599-7264, USA. [5] Research Institute of Biological Psychiatry, Mental Health Center Sct. Hans, Roskilde 4000, Denmark. [6] Department of Psychiatry and Neuroscience, Icahn School of Medicine at Mount Sinai, New York, NY 10029, USA. [7] Department of Genetics and Genomic Sciences and Institute for Genomics and Multiscale Biology, Icahn School of Medicine at Mount Sinai, New York, NY 10029, USA. [8] Mental Illness Research Education and Clinical Center (MIRECC), James J. Peters VA Medical Center, Bronx, NY 10468, USA. [9] MIRECC, JJ Peters VA Medical Center, Bronx, NY 10468, USA. [10] Department of Psychiatry and Human Behavior, Center for Psychiatric Neuroscience, University of Mississippi Medical Center, Jackson, MS 39216, USA. [11] Department of Biology, Duke University, Durham, NC 27708, USA. [12] Department of Human Genetics, University of Chicago, Chicago, IL 60637, USA. [13] Department of Psychiatry, SUNY Upstate Medical University, Syracuse, NY 13210, USA. [14] Department of Biostatistics and Bioinformatics, Duke University, Durham, NC 27708, USA. [15] Department of Medicine, Duke University, Durham, NC 27708, USA. [16] Department of Psychiatry, University of North Carolina, Chapel Hill, NC 27599-7264, USA. [17] Department of Pediatrics, Division of Medical Genetics, Duke University, Durham, NC 27708, USA. These authors contributed equally: Julien Bryois, Melanie E. Garrett, Lingyun Song. Correspondence and requests for materials should be addressed to P.F.S. (email: pfsulliv@med.unc.edu) or to G.E.C. (email: greg.crawford@duke.edu)

Schizophrenia genomics is progressing rapidly, and our mechanistic understanding of this common and often devastating neuropsychiatric disorder is markedly better than 5 years ago[1]. Evidence for a non-specific genetic component for schizophrenia has been known for decades (e.g., sibling recurrence risk of 8.6 and phenotype heritability estimates of at least 60%)[2,3]. The bulk of the genetic basis of schizophrenia is due to common variation[4]. A 2014 paper identified 108 genetic regions and a subsequent report has added over 40 new regions, but the implicated regions are broad and usually do not implicate specific genes[4–6]. It was hypothesized that schizophrenia risk would include many exonic variants of strong effect, but subsequent large whole-exome sequencing studies provide minimal support for this hypothesis[4]. Identifying actionable genes has proven complex, with a few exceptions such as rare exon variants in *SETD1A*[7] and copy number variation in single genes like *NRXN1* and *C4*[8,9].

These studies provide strong evidence that genetic risk for schizophrenia results from the concerted effects of many genes. Schizophrenia may be a disorder of subtly altered amounts of protein isoforms rather than changes in individual amino acids. Non-coding regulatory variation is a major contributor to risk for schizophrenia[4,5], and genomic regions associated with schizophrenia are enriched for gene expression quantitative trait loci (eQTLs) identified in human brains[10,11]. Combining functional genomic data with genome-wide association (GWA) results may be crucial to deciphering connections to specific genes and disease mechanisms in schizophrenia[11–13]. ENCODE[14] and the Roadmap Epigenomics Mapping Consortium[15] provided considerable human functional genomic data and insights into genomic function. However, these studies provided limited insight into psychiatric disorders as most samples were non-neuronal and none were from affected individuals. This study is part of the PsychENCODE consortium which intends to provide functional genomic data from the brains of individuals with and without severe neuropsychiatric disorders[16].

Epigenetic changes in the brain are widely hypothesized to partly mediate risk for schizophrenia[17]. Multiple epigenetic changes have been assessed in schizophrenia (reviewed in ref. [18]), and methylation differences in peripheral blood[19] and brain[20] have been associated with schizophrenia. Of the many epigenetic changes, chromatin accessibility is particularly important and is a conserved eukaryotic feature characteristic of active regulatory elements, including promoters, enhancers, silencers, insulators, transcription factor binding sites, and active histone modifications[21]. Chromatin accessibility has not been systematically evaluated in human brain for schizophrenia, except for a small study[22]. Preferentially accessible regions of chromatin can readily be identified by high-throughput sequencing following the transposition of sequencing adaptors into the DNA backbone via the Tn5 transposase using the assay for transposase-accessible chromatin sequencing (ATAC-seq)[23]. Unlike other nuclease-sensitivity assays, this approach is amenable to limited amounts of postmortem tissue. As chromatin accessibility can differ by >30% between tissues and cell types[24], it is important to study the brains of schizophrenia cases and controls.

Our overall goal was to comprehensively identify active gene regulatory elements in a brain region relevant to schizophrenia, and to quantify how genetic variation alters function such as single-nucleotide polymorphisms (SNPs) that alter chromatin accessibility (i.e., chromatin QTL (cQTL)). We use these data to parse genetic risk for schizophrenia from large GWA. These analyses provide insight into the molecular mechanisms governing schizophrenia risk. To our knowledge, this is the largest study of chromatin accessibility in schizophrenia and among the largest for any human disease.

## Results

**Overview.** In the Methods section, we provide the rationale for our choices of ATAC-seq[23], brain region, and study design. Key features of our approach include the use of the same samples subjected to messenger RNA-sequencing (mRNA-seq) and genotyping analysis by the CommonMind Consortium[11], as well as careful experimentation, including randomization, blinding, comprehensive quality control, empirical covariate selection, and verification of subject identity.

We performed ATAC-seq on 314 brain samples (142 schizophrenia, 143 control, 23 mood disorders, and 6 other). After quality control, the analysis dataset consisted of ATAC-seq on postmortem Brodmann area 9 (dorsolateral prefrontal cortex (DLPFC)) tissue from 135 cases with schizophrenia and 137 controls (Fig. 1a). Sixteen individuals with mood disorders were included for open chromatin peak calling and cQTL analyses, but otherwise excluded. Table 1 summarizes the demographic and clinical features of the subjects. Cases and controls were comparable for sex, ethnicity, age at death, and postmortem brain pH. Cases had greater postmortem intervals (PMIs) and lower RNA integrity number (RIN) scores relative to controls. These differences appeared to have a lesser impact on DNA-based ATAC-seq assays as cases were comparable to controls for unique aligned reads and normalized open chromatin peak calls. However, these differences motivated comprehensive and careful selection of covariates (see Methods).

**ATAC-seq evaluation.** ATAC-seq data were aligned and peaks were called using MACS2 (false discovery ratio (FDR) <0.01, Fig. 1b and Supplementary Fig. 1a–d). We performed extensive quality control (see Methods) to ensure that our final peak set accurately represented the peaks detected in individual samples (Supplementary Fig. 1c–h), that the samples were enriched in our final peak set (Supplementary Fig. 2a, b), and that our peak set was of comparable quality to ATAC-seq from other tissues and from sorted brain nuclei (Supplementary Fig. 3). We identified 118,152 open chromatin peaks totaling 35.5 Mb. As a crude comparison, the human exome has around 131 K exons totaling 47 Mb. We compared these regions of open chromatin to those from smaller experiments (Supplementary Table 1). Allowing for small sample sizes, overlap of our open chromatin results with that identified in these other studies was greatest for the most similar studies (DLPFC in adults), somewhat lower in adult cortical samples of sorted neurons, and lower in fetal cortex. Overlap with the diverse ENCODE samples was low, but higher for CNS-relevant samples. These results show congruence of our larger ATAC-seq data with prior experiments, and underscore the need to study brain.

About a quarter of open chromatin regions map near a protein-coding transcription start site (TSS): 23% were ±5 kb, and 53.0% were >25 kb from any TSS (44% of the latter located downstream of the closest TSS, 24% located within the gene body and 32% located upstream of the closest gene). This distribution was similar to previous studies using DNase-seq (13% at TSS ±2 kb, 26% within the gene body and 34% intergenic)[24]. There was a stronger enrichment for ATAC-seq reads at the TSS for genes that are highly expressed compared to genes that were not expressed (Fig. 1c). We compared the 118,152 peaks to putative regulatory elements from 101 cell types (excluding brain-related regions) from reg2map as part of the Epigenome Roadmap project. We found that 84% of our brain ATAC-seq peaks overlapped a promoter or enhancer in one or more of these cell types (Fig. 1d). This indicates that 16% of the ATAC-seq peak calls are unique to the brain frontal cortex and highlights the need to identify putative regulatory elements from disease-relevant

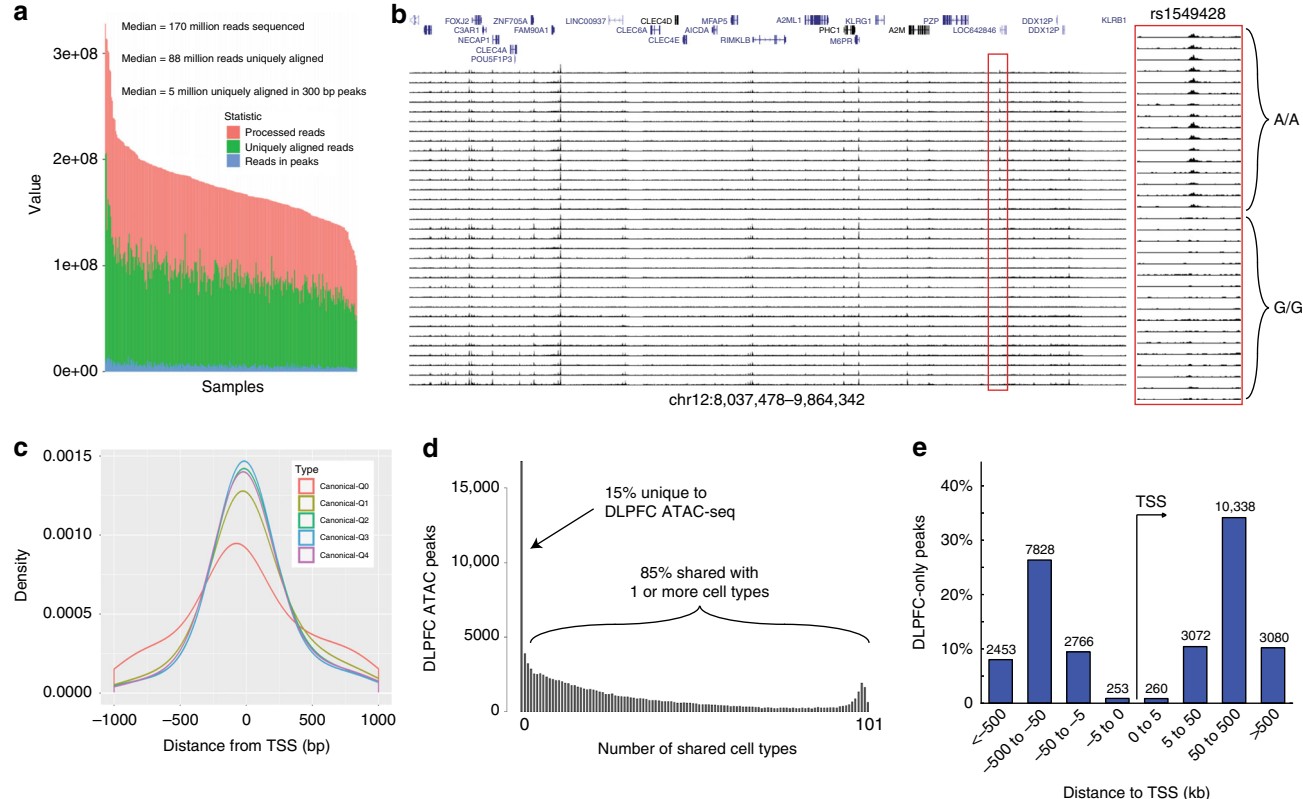

**Fig. 1** ATAC-seq on frozen DLPFC samples. **a** Sequencing statistics from all libraries ($N = 288$) show largely similar total number of reads (pink), uniquely aligning reads (green), and reads that map to ATAC-seq peaks (blue). **b** Individual brain ATAC-seq data in a representative genomic region showing largely congruent identification of regions of open chromatin. Some samples have lower signal to noise (correlated with covariates like postmortem interval and RNA integrity number). Region in red is a chromatin QTL, note lower signal for individuals with GG genotype vs. AA genotype. **c** Open chromatin in relation to transcription start sites (TSS). The TSS is at zero, right is inward in the direction of transcription, and left is outward from the gene. Density curves for all known GENCODE v25 transcripts (one principal transcript per protein-coding gene). Colored curves show different expression levels in DLPFC: Q0 = no expression, Q1–Q4 = lowest to highest expression quartiles. **d** Number of DLPFC ATAC-seq peaks that overlap with putative regulatory elements identified from 101 different cell types analyzed by the Epigenome Roadmap Project (brain tissues excluded). Approximately 16% ($n = \sim 17{,}000$) ATAC-seq peaks are unique to DLPFC. **e** Location of DLPFC-only ATAC-seq peaks relative to the TSS indicates that the majority are in non-promoter regions

### Table 1 Sample description

| Variable | Cases | Controls | Comparison |
|---|---|---|---|
| Subjects after quality control | 135 | 137 | N/A |
| Male sex, N (%) | 92 (68.2%) | 73 (53.3%) | $\chi^2_1 = 6.30$, $P = 0.012$ |
| European ethnicity, N (%)[a] | 112 (83.0%) | 98 (71.5) | $\chi^2_3 = 7.92$, $P = 0.048$ |
| Age at death, mean (SD) | 73.3 (12.6) | 73.9 (17.7) | $F_{1,271} = 0.09$, $P = 0.76$ |
| Postmortem brain pH, mean (SD) | 6.47 (0.24) | 6.50 (0.25) | $F_{1,234} = 0.88$, $P = 0.35$ |
| Postmortem brain mass (g), mean (SD) | 1207 (173) | 1155 (166) | $F_{1,269} = 6.46$, $P = 0.012$ |
| Postmortem interval (h), mean (SD) | 24.3 (15.7) | 10.9 (7.6) | $F_{1,270} = 80.2$, $P < 0.0001$ |
| RNA integrity number, mean (SD) | 7.11 (0.79) | 7.57 (0.84) | $F_{1,270} = 21.5$, $P < 0.0001$ |
| Unique aligned reads (×10⁶), mean (SD) | 89.12 (17.0) | 89.80 (16.1) | $F_{1,270} = 0.12$, $P = 0.73$ |
| Normalized peak calls (FDR 0.01), mean (SD) | 117.3 (70.1) | 130.4 (61.6) | $F_{1,270} = 2.66$, $P = 0.10$ |

All samples were from Brodmann area 9 of left hemisphere
[a]Additional ethnicities in cases were African American (17, 12.6%), Hispanic (5, 3.7%), and Asian (1, 0.74%), and in controls African American (20, 14.6%), Hispanic (16, 11.7%), and Asian (3, 2.2%). The RNA-based measure is pertinent for the DNA-based ATAC-seq assay as the samples were from the same aliquots

tissues. Most of these DLPFC-unique ATAC-seq peaks did not map to promoter regions (Fig. 1e). We did not observe a significant enrichment in the number of ATAC-seq peaks at schizophrenia loci as approximately 1% of our ATAC-seq peaks were located at schizophrenia loci, which cover approximately 1% of the genome.

Our ATAC-seq peaks were enriched in the motifs of a large number of transcription factors, indicating that these data capture

functionally relevant regulatory elements. Notably, we found the strongest enrichment for the following motifs: *CTCF*, *MEF2A*, *MEF2D*, *SP1*, *Zfp410*, *KLF5*, *NFIX*, *JUND*, *ASCL1*, *bhlha15*, and *ZIC4* (all with MEME *e* value $< 10^{-50}$).

Altogether, our evaluation indicates that our ATAC-seq peaks have a relatively similar quality to ATAC-seq data from other tissues, that their locations relative to the closest TSS are relatively similar to open chromatin regions from other tissues, that some

peaks are uniquely found in the DLPFC and that ATAC-seq peaks are enriched in the motifs of transcription factor binding sites, indicating that they capture functional regulatory elements.

**ATAC-seq peaks are enriched for schizophrenia heritability.** We evaluated the relevance of these DLFPC open chromatin regions to schizophrenia using partitioned linkage disequilbrium (LD) score regression[25]. This method evaluates whether the SNP heritability of schizophrenia is enriched in pre-defined genomic features. This approach accounts for multiple technical issues, and conducts a head-to-head comparative evaluation of dozens of genomic features. An earlier analysis found that the SNP heritability of schizophrenia was strongly enriched in evolutionarily conserved genomic regions, but not in open chromatin regions from non-brain cell lines or tissues[25]. We tested heritability enrichment using the same genomic features including our DLPFC ATAC-seq peaks (see Methods)[25].

We found that regions of open chromatin in adult DLPFC were strongly enriched for genetic variation relevant for schizophrenia (Fig. 2a): the 1.2% of SNPs ($n = 125{,}762$) located in ATAC-seq

peaks explained 8.55% of the SNP heritability of schizophrenia (7.1-fold enrichment, $P$ value = 0.015). This level of enrichment was close to the level of enrichment of conserved regions[25]. We replicated this result in an independent ATAC-seq dataset from the Chicago psychENCODE group (UIC) based on 265 adult DLPFC samples from an independent cohort processed with the same bioinformatics pipeline. We obtained strikingly similar results with 1.6% of the SNPs located in this new set of peaks explaining 9.6% of the SNP heritability of schizophrenia (5.9-fold enrichment, $P$ value = 0.026, Supplementary Fig. 4). Thus, common genetic variation that mediates risk for schizophrenia is not randomly distributed in the genome, but is concentrated in definable genomic features, particularly regions of open chromatin in the brain cortex.

To evaluate the specificity of this result, we compared open chromatin from DLPFC to that in 138 cell types and tissues generated by the ENCODE Consortium using DNase-seq, or by us and other groups using ATAC-seq. The DLPFC ATAC-seq data displayed the greatest association with schizophrenia compared to any other cell or tissue type tested (Fig. 2b and Supplementary Fig. 5). These data again indicate the importance

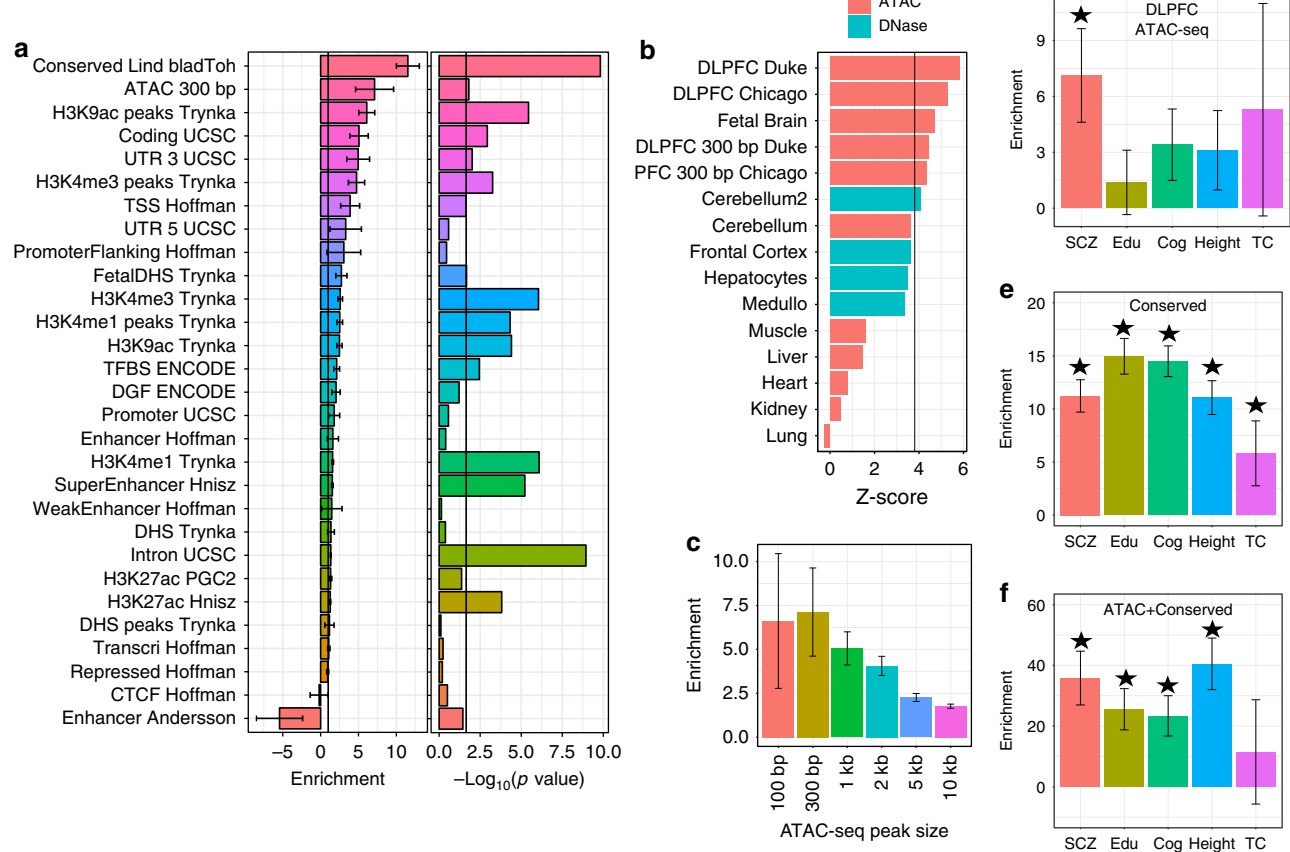

**Fig. 2** Schizophrenia heritability is enriched for brain-specific accessible chromatin. **a** Schizophrenia heritability enrichment (standard error) and significance level (–log 10(*P*)) of different functional genomic annotations estimated using partitioned LD score regression. Enrichment of ATAC-seq regions in DLPFC is second only to genomic regions conserved across 29 Eutherian mammals. The black bar represents the 5% false discovery rate threshold. **b** Enrichment across a subset of 142 DNase-seq and ATAC-seq datasets (see Supplementary Fig. 9 for complete comparison). The black bar represents the Bonferroni significance threshold (=0.01/142). Top enrichments are in DLPFC ATAC-seq peaks generated from Duke and U Chicago groups, followed by other mostly brain-specific tissues and cell lines. ATAC-seq tissue samples (cerebellum, liver, muscle, heart, kidney, and lung) represent an independent study to control for batch and method effects. **c** Schizophrenia heritability enrichment and standard error for peaks of different width. **d** Heritability enrichment of ATAC-seq peaks from brain display significant enrichment (*$P < 0.05$) for GWA variants associated with schizophrenia, but not for educational attainment (Edu), cognitive ability (Cog), height, and total cholesterol (TC). **e** Heritability enrichment of evolutionarily conserved regions are significantly enriched for schizophrenia, educational attainment, cognitive ability and height, and TC. **f** Heritability is ~4× more enriched for regions that overlap between ATAC-seq and conservation than for regions that are either conserved or in ATAC-seq peaks

of studying relevant tissues from cohort samples. DNase-seq from frontal cortex, cerebellum, a neuroblastoma cell line, and a medulloblastoma cell line also showed more association relative to samples not from the brain (Supplementary Fig. 5). We also generated ATAC-seq from fetal brain samples that showed slightly lower but still significant enrichment, indicating that relevant regulatory regions that confer risk are accessible earlier in development (Fig. 2b).

To test whether SNPs nearer to the peak centers were more likely to be causal, we determined heritability enrichment of ATAC-seq peaks of varying widths (100 bp, 300 bp, 1 kb, 2 kb, 5 kb, and 10 kb). Schizophrenia SNP-heritability enrichment decreased as peak width increased (Fig. 2c), indicating that SNPs nearer to the peak center explained more SNP heritability than SNPs further away.

The SNP-heritability enrichment of DLPFC ATAC-seq peaks was specific to schizophrenia, and was not significantly enriched for GWA variants for educational attainment, cognitive ability, height, or total cholesterol (Fig. 2d). This is in contrast to evolutionarily conserved regions, which are enriched for GWA variants for schizophrenia, educational attainment, cognitive ability, height, and total cholesterol (Fig. 2e). To investigate whether sub-regions of the ATAC-seq peaks were particularly enriched for schizophrenia SNP heritability, we intersected the ATAC-seq peaks with evolutionarily conserved regions[26], and found that conserved regions located in DLPFC ATAC-seq peaks were extremely enriched in SNP heritability (36-fold enrichment, $P$ value $= 8 \times 10^{-7}$, Fig. 2f). Conserved regions that map within ATAC-seq peaks cover 6 Mb, which is much smaller than the 35 Mb covered by ATAC-seq peaks. The intersection of open chromatin and conserved regions tended to be near protein-coding TSS with 41% located ±10 kb from the TSS of the closest protein-coding gene, whereas only 30.9% of the ATAC-seq peaks and 25.3% of the conserved regions were located ±10 kb of a TSS. Our results support that DLPFC ATAC-seq peaks are enriched for risk variants in schizophrenia, and that restricting these regions to those that are evolutionarily conserved will allow the fine mapping of schizophrenia risk loci.

Conserved regions in ATAC-seq peaks were significantly enriched for *CTCF* binding sites (background: all ATAC-seq peaks, Homer $P$ value $<1 \times 10^{-100}$, MEME-chip $e$ value $<1 \times 10^{-100}$), which are involved in the formation of topologically associated domains[27]. In addition, these regions were strongly enriched for the motif of *RFX1*, a gene implicated in the regulation of neuronal glutamate transporter type 3[28]. (Supplementary Table 2). This indicates that common genetic variation implicated in schizophrenia could impact higher-order DNA conformation and/or affect neuronal functions.

**Identification of differential open chromatin**. We then aimed to characterize the effect of age at death, PMI and schizophrenia on chromatin accessibility in the prefrontal cortex (see Methods). For age at death, we detected 2310 peaks showing significant differences (5% FDR) in chromatin accessibility as a function of age (Fig. 3a, b and Supplementary Data 1). Genes located in close proximity to age affected peaks (9396 peaks, 20% FDR) were significantly enriched in terms related to cell differentiation, oligodendrocyte specification, and neuron differentiation (Supplementary Data 2). Peaks that became more open with age (20% FDR) were enriched for motifs of several transcription factors (*Foxj3*, *FOXC2*, *ZNF263*, *Zfp281*, *SP1*, *FOSL1*, *JUND*, *JUNB*, *Tcf12*, *ASCL1*, *Myog*, *SOX10*, *Mrf1*, *Sox11*), while peaks that became more closed with age were enriched for other transcription factors (*SP2*, *Zfx*, *Zic1*, *NFIX*, *NFIA*, *NFIB*, *ZNF263*, *Sp4*, *POU6F2*, *Pou6f1*, *Sox21*). Interestingly, *ASCL1*, *SOX10*, *Sox11*, *SP2*, *Zic1*, *NFIX*, and

*Sox21* are all implicated in neurogenesis[29–35]; *NFIA* plays an important role in oligodendrocyte maturation[36], while *NFIB* plays an important role in neural progenitor self-renewal[37]. In addition, *Sp4* and *Pou6f1* were shown to regulate dendritic patterning[38,39]. Altogether, these results indicate that differentially accessible chromatin regions with age capture genes and transcription factors implicated in neuronal development and might also reflect a change in cell heterogeneity with age (e.g., fewer neurons).

For PMI, we detected 466 peaks showing significant differences in chromatin accessibility (Fig. 3c, d and Supplementary Data 3). Genes located in close proximity of peaks affected by PMI (2328 peaks, 20% FDR) were significantly enriched in a large number of biological functions, including the p53 pathway, the insulin/insulin-like growth factor pathway, hypoxia response, and tumor growth factor-β receptor binding (Supplementary Data 4). Peaks that displayed increased accessibility with PMI (20% FDR) were enriched for multiple transcription factors (*SRF*, *Tbp*, *Bbx*), while peaks with decreased accessibility were enriched for other transcription factor motifs (*PLAG1*, *ZNF263*, *EWSR1-FLI1*, *Zfp281*, *ZNF384*, *Mtf1*, *Foxl1*).

For schizophrenia, we detected three regions differentially accessible between cases and controls (Fig. 3e, f, Supplementary Fig. 6, and Supplementary Data 5). We replicated the association of our top hit (chr2:132,130,366–132,130,666, $P$ value $= 6.6 \times 10^{-5}$, same direction of effect) in the UIC dataset (15 cases/170 controls with enrichment in our peaks >2×, see Methods), while our second and third hits did not replicate ($P$ value $= 0.96$ and $P$ value $= 0.72$, respectively). The 1000 peaks with highest evidence of being differentially accessible between cases and controls were located close to genes significantly enriched in functions related to vitamin B6 metabolism and L-carnitine biosynthesis (enrichment $= 11 \times -17 \times$, $q$ value $= 0.03-0.08$, Supplementary Data 6). In order to test whether these genes were enriched in genetic association with schizophrenia, we performed a gene-set enrichment analysis using MAGMA[40]. We found that they were not significantly associated with schizophrenia genetics (MAGMA $P$ value $= 0.97$), suggesting that, on average, these genes are not playing a causal role in schizophrenia. This indicates that the biological enrichments in vitamin B6 and L-carnitine biosynthesis are unlikely to be causal but more likely to reflect consequences of the disorder or the effect of unaccounted confounders (difference in lifestyle, medication, etc.) between cases and controls.

More accessible peaks in cases (top 1000 peaks) were enriched for the motifs of multiple transcription factors (*ALX4*, *POU5F1P1*, *ZNF263*, *SP2*, *E2F6*, *Nr5a2*), while less accessible peaks were enriched in the motifs of *ASCL1*, *ZEB1*, *RXRG*, *Zfp281*, *RXRA*, *PLAG1*, and *GATA1*. Interestingly, *Sp2*, *Nr5a2*, *ASCL1*, and *ZEB1* are all implicated in neurogenesis[31,41–43], while *RXRG* is implicated in myelin regeneration[44].

A recent publications performed RNA-seq on the same samples and discovered 693 genes differentially expressed (5% FDR) between cases and controls[11]. We observed that 4613 ATAC-seq peaks were located nearby these 693 genes (gene coordinates extended by 30 kb upstream to 10 kb downstream). We found no evidence that peaks in these differentially expressed genes were differentially accessible between cases and controls (top $q$ value $= 0.66$). In addition, peaks in these genes did not have significantly lower differential accessibility $P$ values than expected by chance ($P = 0.5$). We also performed a differential chromatin analysis between cases and controls based on the promoter region of each gene (2 kb upstream to TSS) and did not observe any significant difference in chromatin accessibility for any gene (top $q$ value $= 0.297$). Several possibilities can explain these results: (1) the biological mechanism leading to differential expression might not depend on changes in chromatin

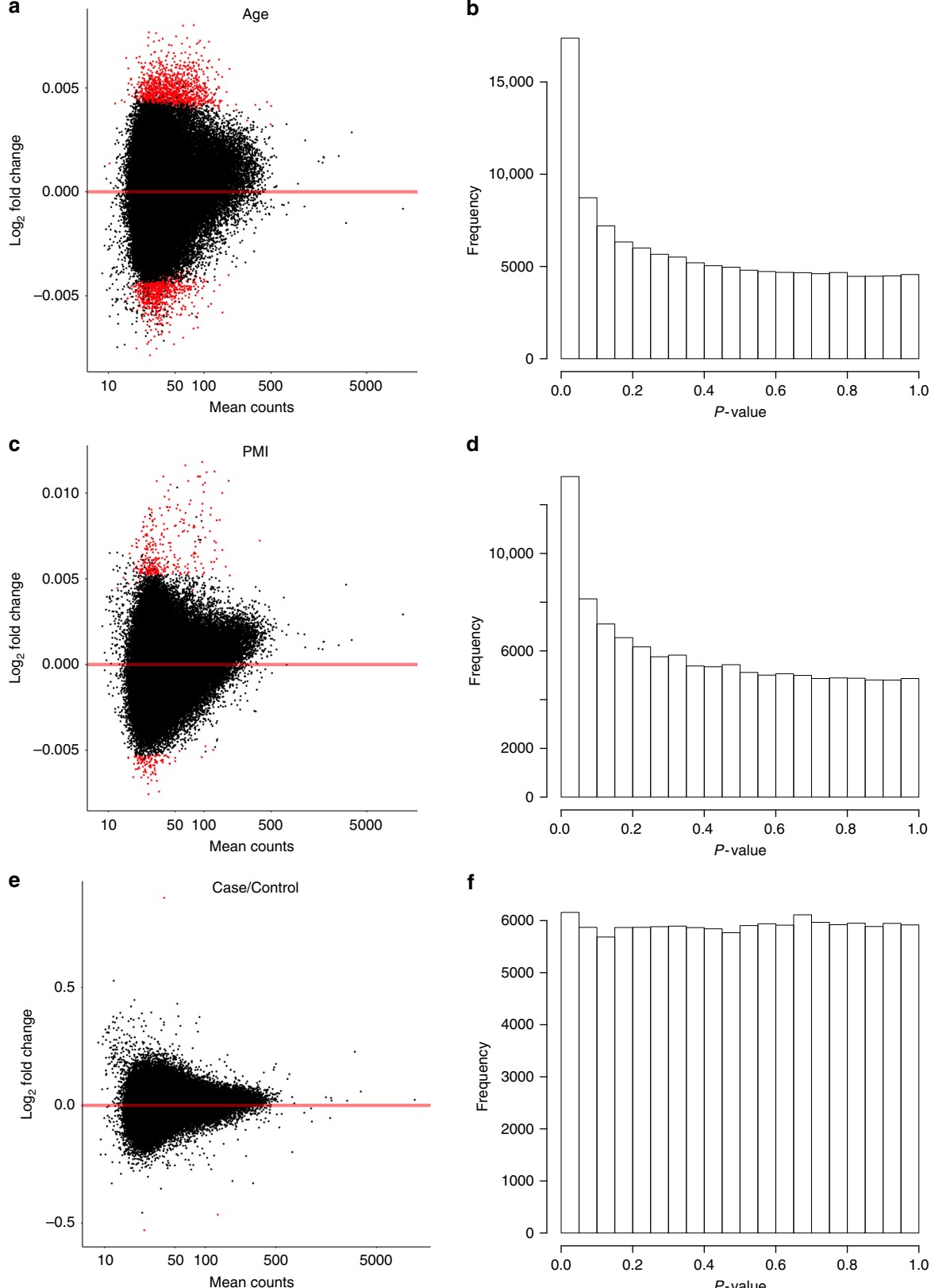

**Fig. 3** Differential chromatin accessibility differences detected in 288 brain samples. **a** Differential chromatin detected as a function of age at death. Dots in red indicate significantly differential regulatory elements (FDR <0.05). **b** Distribution of *P* values for age. **c** Differential chromatin detected as a function of postmortem interval (PMI). **d** Distribution of *P* values for PMI. **e** Differential chromatin detected as a function of case–control status. **f** Distribution of *P* values for case–control status. Peaks on chrX were meta-analyzed (inverse variance weighted) and peaks on chrY were only tested in males

 

accessibility, (2) there might be lower statistical power to find differential chromatin accessibility than differential expression, and (3) cell-type tissue heterogeneity could mask differential chromatin.

Altogether, our results indicate that differences in chromatin accessibility between cases and controls in postmortem human brain prefrontal cortex are relatively minor in comparison with changes in chromatin accessibility due to age or postmortem interval.

**Identification of cQTLs.** Previous studies showed that schizophrenia risk alleles are enriched for brain eQTLs[10]. To determine if genetic variants associated with schizophrenia can also impact chromatin accessibility, we performed a cQTL analysis using our DLPFC ATAC-seq data (Fig. 4, see Methods). In total, we identified 6200 SNPs that were significantly associated with differences in chromatin accessibility (5% FDR, Fig. 4a). There was no skewing of rare or common allele frequency in this subset of SNPs (Fig. 4b). Of these cQTLs, 622 (10%) were located in the chromatin peak with which they were associated, and the majority of cQTLs (52.0%) were ±2 kb from the center of the peak with which they were associated, indicating that most cQTLs act locally (Fig. 4b). The most significant cQTL was rs1549428 associated with an open chromatin peak at chr12:9,436,157–9,436,457 ($q = 3.9 \times 10^{-82}$, Figs. 1b, 4c). We observed that 176 of the 6200 cQTLs (2.8%) were located at a

schizophrenia GWA locus[5], the most significant of which was rs11615998 in *CACNA1C* ($q = 2.7 \times 10^{-44}$, Fig. 4d). As an independent confirmation, we show that individuals who are heterozygous for cQTL alleles display allele skewing (Fig. 5a, b) in the direction that is expected (Fig. 5c). Similar to eQTLs[11], we detected no significant enrichment of cQTLs in schizophrenia GWA regions.

**cQTLs overlap with eQTLs.** We compared cQTLs to previously identified eQTLs in the same dataset[11]. The estimated proportion of significant cQTLs that are also eQTLs is 23.3% (see Methods), which agrees with similar estimates from lymphoblastoid cell lines (23%)[45]. For SNPs that were both cQTLs and eQTLs, 63.6% exhibited concordant effects ($P < 1 \times 10^{-4}$), meaning the allelic association indicated both more open chromatin and higher gene expression (Fig. 6a). Despite this correlation of direction of the effect, there was only a weak correlation in the sizes of the effect ($r = 0.21$) (Fig. 6b). Finally, for SNPs in high LD, the direction of effect for cQTL and eQTL was not always the same (Fig. 6c, d). Looking at the direction for all QTLs that have pairwise SNPs with $r^2 > 0.8$, approximately 31% go in the opposite direction. This is roughly the same percentage going in the opposite direction as it is among all QTLs, regardless of LD (Fig. 6a). This may indicate independent mechanisms by which these SNPs impact gene expression.

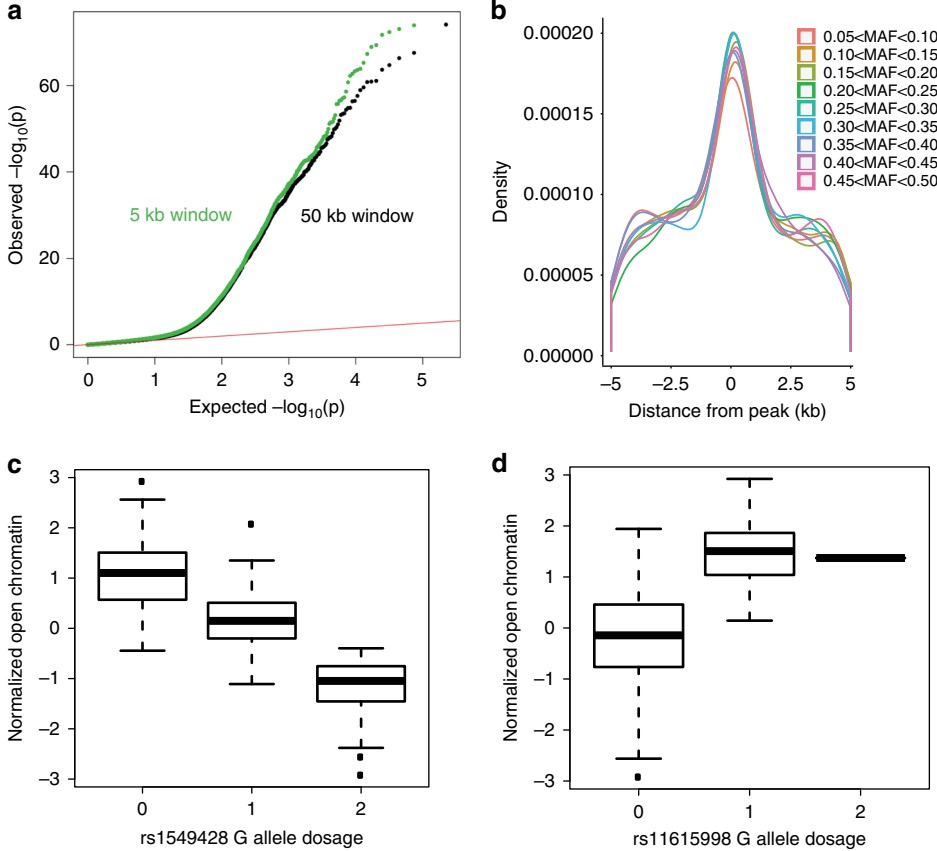

**Fig. 4** Identification of chromatin QTLs (cQTLs). **a** QQ plot of cQTLs using window size of 5 kb (green) or 50 kb (black). Both show marked deviations from the expected. **b** Distance of cQTLs relative to the center of ATAC-seq peaks as a function of minor allele frequency. **c** Most significant cQTL, rs1549428 (chr12:9,436,157–9,436,457); individuals homozygous for the reference allele (0) display more chromatin accessibility than individuals that are heterozygous (1) and homozygous for the alternate allele (2). **d** Most significant cQTL that is also a schizophrenia GWA loci, rs11615998 (chr12:2,364,960 −2,365,260). Due to a low minor allele frequency (MAF = 0.059), only one individual was homozygous for the alternate allele. The results were comparable with or without this homozygous individual. The boxplots represent the following statistics: median (bolded line), the 1st and 3rd quartiles (bounds of box) and 1.5× the inter-quartile range (whiskers)

 

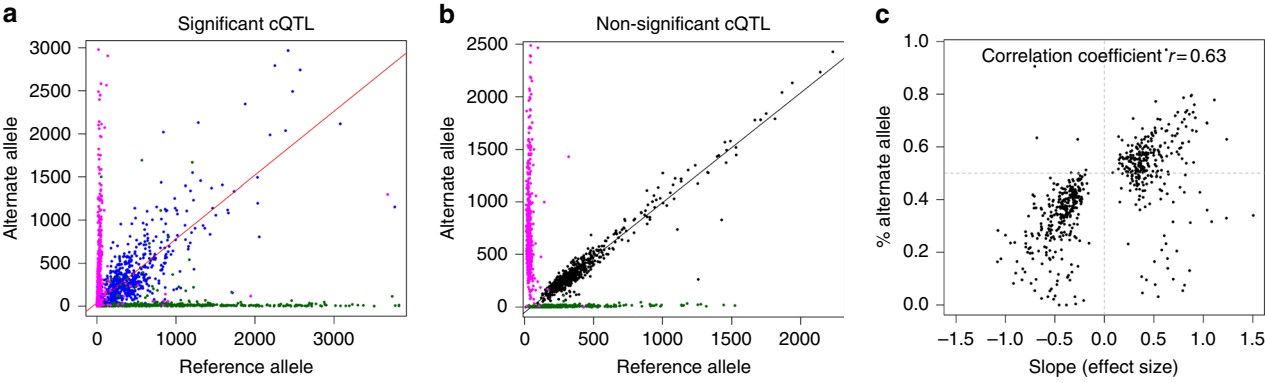

**Fig. 5** Heterozygous individuals for cQTLs display allele bias. **a** Allele counts across cQTLs for individuals homozygous for reference alleles (green), homozygous for alternative alleles (magenta), or heterozygous for cQTL variants (blue). **b** Same as **a**, but for non-significant cQTLs. **c** For individuals heterozygous for cQTLs, we determined the % of each cQTL that correlated to the alternate allele (*y*-axis) and compared those values to the cQTL effect size (*x*-axis)

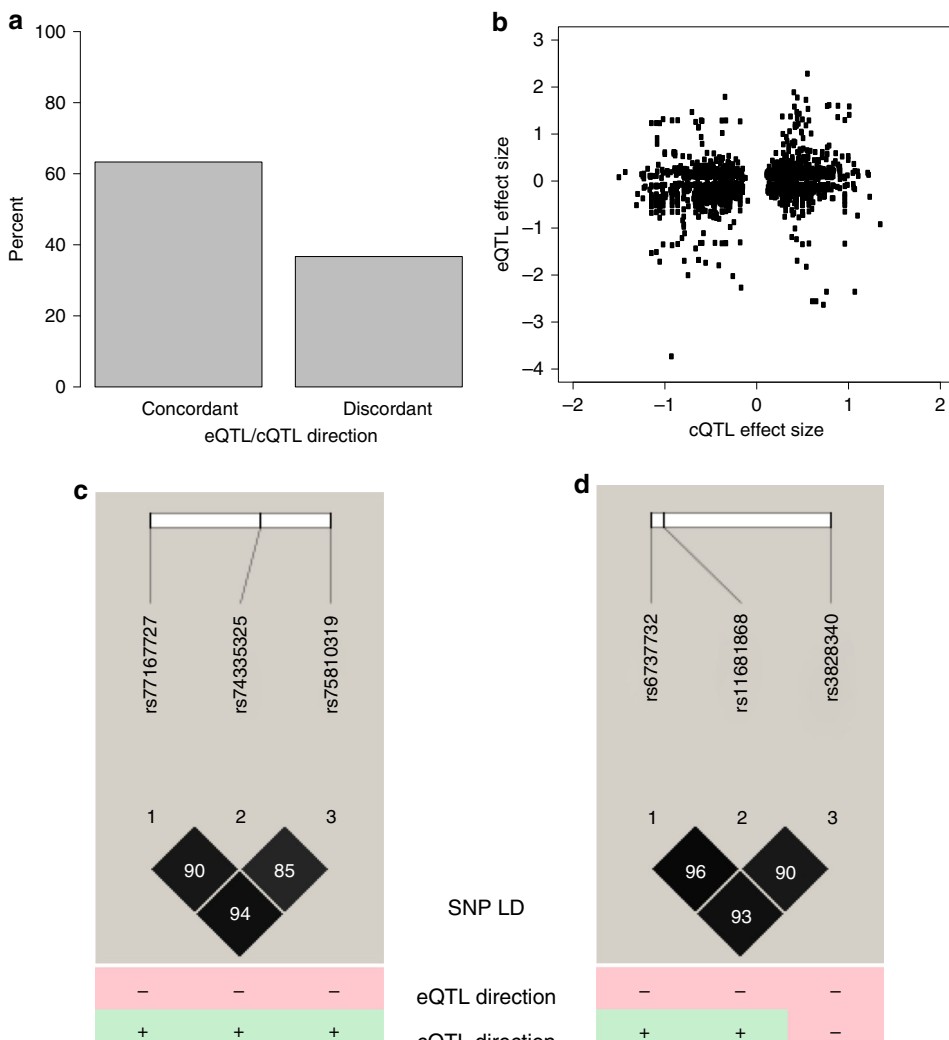

**Fig. 6** SNPs that are both eQTLs and cQTLs. **a** Direction of effects of cQTL vs. eQTL. Concordant effects are when cQTL allele associated with greater open chromatin is also associated with higher eQTL expression. **b** Even though the direction of cQTL and eQTL is largely concordant, the effect sizes of these differences are only modestly correlated (*r* = 0.21). **c** Three SNPs in high LD on chromosome 1. The tested allele is associated with lower expression and more open chromatin for all three SNPs. Distance between SNPs 1 and 2 is 10 kb, and distance between SNPs 2 and 3 is 6.3 kb. **d** Three SNPs in high LD on chromosome 2. The tested allele is associated with lower expression in all three instances, but with more open chromatin for two SNPs and more closed chromatin for the third SNP. Distance between SNPs 1 and 2 is 2 kb and distance between SNPs 2 and 3 is 20 kb

**cQTLs colocalize with schizophrenia GWAS regions**. Finally, we performed a Bayesian colocalization analysis to identify regions with high probability of a shared causal variant associating both with schizophrenia and chromatin accessibility. Using a posterior probability cutoff of 0.9, we identified eight regions with evidence for a shared genetic effect (Table 2). Two separate cQTLs in *MAD1L1* (PP(H$_4$) > 0.98), one cQTL in *AS3MT* (PP(H$_4$) = 0.98), one cQTL in *TSNARE1* (PP(H$_4$) = 0.94), and one cQTL in *RERE* (PP(H$_4$) = 0.92) were found to colocalize with schizophrenia-associated variants. Two cQTLs that colocalized with schizophrenia lie in intergenic regions on chromosomes 16 and 1. Finally, one colocalized peak lies in the pseudogene *GOLGA6L5P*. We also performed colocalization analysis between cQTLs and eQTLs in those eight regions for all genes with a significant eQTL. We observed that only the cQTL located in the AS3MT gene colocalized with eQTLs (eQTLs of both *AS3MT* and *WBP1L*), indicating that a single genetic variant can affect the expression level of multiple genes at schizophrenia GWAS loci. To further characterize these colocalized regions near *AS3MT*, we utilized chromatin interaction data from eHi-C analysis to examine the extent of chromatin looping in the colocalized region (Supplementary Fig. 7). Qualitatively, it appears that there are many chromatin interactions involving regions of shared genetic effect between schizophrenia, DLPFC gene expression, and open chromatin. Thus, while these data

provide important functional context for the schizophrenia GWAS associations, more sophisticated functional analysis will be needed to dissect the specific mechanisms leading to schizophrenia risk.

**Discussion**

Groups such as the Psychiatric Genomics Consortium have generated some of the largest GWA data for schizophrenia to date[1]. These studies have largely pointed to non-coding regions of the genome, suggesting that variants in gene regulatory elements contribute to schizophrenia risk. This is supported by whole-exome sequencing efforts that have identified few pathogenic coding variants[7], as well as expression data showing that brain eQTLs are enriched for schizophrenia association statistics[11]. Thus, we believe that the next step in schizophrenia research is to provide more targeted evidence that schizophrenia risk alleles indeed fall within putative gene regulatory elements, and to characterize the mechanism of those non-coding variants and how they alter gene expression levels. As part of the psychENCODE project[16], we have made progress in both areas, and summarize our results in Table 3.

Our study shows that, of diverse genomic and epigenomic datasets that span many different cell types and tissues, ATAC-seq data from DLPFC is the most associated with schizophrenia

---

**Table 2 Summary of colocalization of cQTL variants with schizophrenia GWA and eQTLs**

| DLPFC ATAC-seq region | Closest gene to DLPFC ATAC-seq region | SCZ GWA region (Ripke et al., 2014) | No. SNPs tested for coloc. with SCZ GWA | Post. prob. of shared genetic variant with SCZ GWA | DLPFC eQTL gene (Fromer et al., 2016) | No. SNPs tested for coloc. with eQTL | Post. prob. of shared genetic variant with eQTL |
|---|---|---|---|---|---|---|---|
| Chr15_85,053,431_85,053,731 | GOLGA6L5P | 29 | 2 | 0.9991 | — | — | — |
| Chr16_58,676,080_58,676,380 | — | 89 | 26 | 0.9951 | — | — | — |
| Chr7_2,103,480_2,103,780 | MAD1L1[a] | 7 | 35 | 0.9865 | — | — | — |
| Chr10_104,629,149_104,629,449 | AS3MT | 3 | 21 | 0.9831 | AS3MT | 17 | 0.986 |
| Chr10_104,629,149_104,629,449 | AS3MT | 3 | 21 | 0.9831 | WBP1L | 17 | 0.915 |
| Chr7_2,030,144_2,030,444 | MAD1L1[a] | 7 | 20 | 0.9804 | — | — | — |
| Chr1_30,427,451_30,427,751 | — | 61 | 15 | 0.9737 | — | — | — |
| Chr8_143,323,367_143,323,667 | TSNARE1 | 5 | 23 | 0.9386 | — | — | — |
| Chr1_8,468,325_8,468,625 | RERE | 53 | 14 | 0.9184 | — | — | — |

[a]Two separate DLPFC ATAC-seq peaks were proximal to *MAD1L1*

---

**Table 3 Summary of results pertaining to schizophrenia genomic findings**

| Empirical findings | Evidence |
|---|---|
| Brain open chromatin regions are significantly enriched for schizophrenia SNP heritability | Fig. 2, Supplementary Fig. 4, Supplementary |
| This finding replicated in an independent sample | Fig. 5 |
| Schizophrenia SNP-heritability enrichment in open chromatin is second to that in regions conserved across 29 Eutherian mammals | Fig. 2a |
| Regions that are both conserved and in open chromatin are particularly enriched | Fig. 2f |
| Regions that are both conserved and in open chromatin are enriched for transcription factor binding sites with neuronal functions | Supplementary Data 2 |
| Many cross-tissue biological features do not show schizophrenia SNP-heritability enrichment (e.g., promoters, other epigenetic marks) | Fig. 2a |
| Schizophrenia SNP-heritability enrichment in the DLPFC is relatively specific and not a feature of GWA generally | Fig. 2d |
| Adult DLPFC displays similar heritability enrichment relative to fetal PFC | Fig. 2b |
| Studying adult brain was crucial: higher enrichment compared to 138 tissues/cell types | Fig. 2b, Supplementary Fig. 5 |
| Few differences in chromatin accessibility between schizophrenia cases and controls. | Fig. 3e, f |
| Six thousand two hundred cQTLs were identified. Eight of them colocalize with schizophrenia GWA signal. One also colocalizes with eQTLs of two genes (AS3MT and WBP1L) | Table 3, Supplementary Fig. 7 |

common variant GWA results. Furthermore, genomic regions conserved across 29 Eutherian mammals within adult DLPFC ATAC-seq peaks display an even higher degree of heritability enrichment. This strongly suggests that multiple orthologous data types will be needed to fine-map risk variants that contribute to schizophrenia. This important discovery likely narrows the search space for identifying risk alleles. Additionally, we have identified eight chromatin peaks with a high probability of shared causal cQTL variant with schizophrenia GWAS variants, one of these peaks also shares genetic effects with an eQTL affecting the expression of *AS3MT* and *WBP1L* in DLPFC. This discovery demonstrates biological mechanisms by which common variants may affect risk for schizophrenia.

We detected three differentially accessible regions between cases and controls. Interestingly, the differences in chromatin accessibility between cases and controls were much less numerous than differences due to age or postmortem interval, indicating that differences in chromatin accessibility in the adult prefrontal cortex between schizophrenia cases and controls are relatively subtle. We also found that genes located in close proximity of the top 1000 peaks with highest evidence of being differentially accessible were likely not causal as they were not significantly enriched for schizophrenia GWA associations. This indicates that the biological enrichment that we observed (vitamin B6 and L-carnitine biosynthesis) for these genes is likely reactive to the disease (e.g., drug treatment) or due to other possible confounders (e.g., difference in lifestyle).

While it appears that differential chromatin accessibility is not pervasive in the adult prefrontal cortex between schizophrenia cases and controls, differential chromatin accessibility may still be involved in the etiology of schizophrenia. The sample size of our study may have been underpowered to identify differences in chromatin state between cases and controls. We also cannot rule out that cell heterogeneity may mask rare but important cell-type-specific signals. In other work, we implicated cortical pyramidal neurons as an important cell type for schizophrenia[46], and as pyramidal neurons comprise ~40% of the DLPFC, this may be less likely. Furthermore, in silico mixing experiments demonstrated that differential chromatin accessibility can accurately be attributed to specific cell types comprising a heterogeneous sample (Supplementary Fig. 8). Another possibility is that differential chromatin accessibility between cases and controls is manifested only within specific developmental windows. Indeed, we have identified chromatin accessibility differences that occur during development in post-mitotic mouse neurons[47]. While careful age-matched comparisons are needed in human to explore this possibility, identifying differences between cases and controls will be challenging as psychotic symptoms typically do not appear until early adulthood.

While these scenarios above are feasible, it is possible that functional non-coding SNPs (including eQTLs) contribute to schizophrenia in a mechanism independent of changes in chromatin accessibility. These functional non-coding variants may still impact specific transcription factor binding sites, but overall chromatin accessibility might be maintained and stabilized by other transcription factors and complexes that bind at these regions. Indeed, conserved sequences that are in brain open chromatin regions are highly enriched in CTCF and other transcription factors involved in neuron differentiation. Altering CTCF binding might also impact three-dimensional genome structure to allow distal enhancers to abnormally influence gene expression[48]. Strategies to explore these possibilities include chromatin immunoprecipitation-sequencing for specific transcription factors, and Hi-C on brains representing different genotypes. Additional independent strategies include functional detection of variants on regulatory element activity using high-

throughput reporter assays like POP-STARR-seq[49] or regulatory element CRISPR screens[50].

Future experiments tackling these issues will be needed if we are to further understand the mechanism behind schizophrenia risk as well as disease risk for many other common disorders. These results can help narrow the search space for those studies.

## Methods

**Study design.** We conducted a case–control comparison of brain samples to investigate the role of chromatin accessibility in the etiology of schizophrenia. To our knowledge, this is one of the largest studies to date of open chromatin in the brain for schizophrenia. The key features were as follows: use of ATAC-seq (a newer method of identifying regions of open chromatin);[23] use of the same brain tissue samples studied with RNA-seq by the CommonMind Consortium;[11] and careful experimental design (including randomization, blinding, comprehensive quality control, empirical selection of salient covariates, and verification of subject identity). We attempted ATAC-seq on 314 brain samples (142 schizophrenia, 143 control, 16 bipolar disorder, 7 affective disorder, and 6 other). After the quality control procedures described below, the analysis dataset consisted of ATAC-seq on postmortem brain samples from 135 cases with schizophrenia and 137 controls. For some analyses (e.g., identification of brain cQTLs, covariate selection, differential chromatin analysis), we included 16 individuals with mood disorders. The purpose of including these individuals was to increase power for the cQTLs detection, to increase power to detect covariates affecting ATAC-seq peaks quantification, and to better estimate parameters that are not dependent on the case–control status in the differential chromatin analysis.

**Rationale: choice of intact brain vs. cell populations.** The brain is a complex mixture of cell types. At present, there is no ideal approach to comprehensively deal with cell heterogeneity. (1) Nuclei sorting from frozen brain provides enrichment for specific cell types[51]. However, sorting is never perfect, relatively few cell types can be sorted in humans, and cells may change state during the 5+ hour process of thawing, dissociation, ultra-centrifugation, and sorting. Sorting neuronal populations from mice with a cell-type-specific fluorescent tag is possible, but there is only an ~50% overlap in regulatory regions between mice and human[52]. (2) Laser capture microdissection on frozen tissue provides spatial resolution, but yields limited quantity and quality of chromatin, and artifacts from thawing and excess heat are concerns. (3) Single-cell analysis of open chromatin is being developed but is technically difficult, resolution is currently limited, and is not yet available for frozen brain samples[53]. (4) Ex vivo stem cell cultures can yield a realistic cell type but all current ex vivo microenvironments do not recapitulate the normal development of the human brain. A further challenge is that the neural cell types that contribute to schizophrenia are not well characterized.

We performed in silico mixing experiments and demonstrated that we can detect cell-type-specific gene regulatory elements at various cell concentrations (Supplementary Fig. 8a). Our analysis of the pros and cons of using intact tissues vs. cell sorting is shown in Supplementary Fig. 8b. As the cell types causally related to schizophrenia are unknown, we believe that generating chromatin maps in intact brain—that is, the union of all cell types present in a brain region associated with schizophrenia—provides one strategy of identifying schizophrenia-relevant regulatory elements. Using intact tissue will allow us to analyze more samples, providing better power to identify chromatin QTLs.

**Rationale: choice of brain region.** DLPFC samples corresponding to Brodmann areas 9 and 46 were studied. These regions were chosen due to their relevance to schizophrenia based on brain anatomy, imaging, and gene expression[54]. DLPFC was also the focus of a recent paper from the CommonMind Consortium[11]. In fact, RNA-seq from that study and ATAC-seq data from this study have been generated on tissue aliquots isolated from the same DLPFC dissections.

**Subjects and brain samples.** We used ATAC-seq to characterize human DLPFC cortical samples from the Mt. Sinai NIH Brain and Tissue Repository (http://icahn.mssm.edu/research/labs/neuropathology-and-brain-banking, Dr. Vahram Haroutunian). All cases met the Diagnostic and Statistical Manual of Mental Disorders, 4th Edition criteria for schizophrenia via standard diagnostic procedures[11]. Controls had never met the criteria for schizophrenia or a psychotic disorder. Subjects were excluded if they had neuropathology related to Alzheimer's or Parkinson's disease, acute perimortem neurological insults, or were on a mechanical ventilator near the time of death. No case or control had a large pathogenic copy number variant, and cases had inherited a significantly greater number of schizophrenia risk alleles[11].

All schizophrenia cases and all controls were dissected from the left hemisphere of fresh-frozen coronal slabs cut at autopsy from the DLPFC corresponding to Brodmann area 9. All bipolar disorder samples were from Brodmann area 46. Immediately after dissection, samples were cooled to −190 °C and dry homogenized to a fine powder using a liquid nitrogen-cooled mortar and pestle. Aliquots from each sample were prepared and used for multiple purposes, including ATAC-seq (reported here), RNA-seq[11], and SNP genotyping (Illumina

OmniExpressExome array). Tissue aliquots were shipped as a dry powder on dry ice to the Crawford lab at Duke University.

Additional adult dorsolateral prefrontal cortex samples (Brodmann area 9, see Supplementary Table 1) were dissected from postmortem samples from nine adult schizophrenia cases and nine adult control brains, and were obtained from Dr. Craig Stockmeier (University of Mississippi Medical Center). Cases and controls were sex and age matched. All adult samples were of European ancestry. Controls had no history of psychiatric disorders or substance abuse.

Frontal cortex from nine fetal brains (Supplementary Table 1), gestation age 17–19 weeks, were obtained from the NIH NeuroBiobank (https://neurobiobank.nih.gov). All fetal samples were of African-American ancestry. Samples were genotyped on the Illumina Human OmniExpress chip in order to confirm sample integrity. Samples were dry homogenized to a fine powder using a liquid nitrogen-cooled mortar and pestle. Aliquots from each sample were prepared and used for multiple purposes, including ATAC-seq, RNA-seq, and DNA microarray. Sample processing was conducted blind to case–control status.

**Confirmation of sample identity**. We confirmed subject identity by comparing Illumina SNP genotypes from the Illumina OmniExpressExome array to those recoverable from the ATAC-seq reads (described below). ATAC-seq reads were aligned to hg19 using bowtie2 and variants were called using the multi-sample HaplotypeCaller according to best practices in the Genome Analysis Toolkit[55]. To achieve a high level of confidence in the variant calls from ATAC-seq, we only kept variants in peaks with a mean read depth ≥10 and with minor allele frequency >0.05. This stringent filtering yielded 10,939 SNPs present in both the ATAC-seq and Illumina data. Identity by descent was then estimated for each pairwise combination of ATAC-seq and GWA samples using PLINK[56]. For two subjects, genotypes from ATAC-seq and Illumina genotyping did not match and were excluded in all further analyses.

**ATAC-seq library preparation and sequencing**. Samples were processed in batches of eight. Samples were randomly assigned to batches that were balanced with respect to case–control status and sex. Sample processing was conducted blind to case–control status. Frozen pulverized brain samples were received from the Mt. Sinai Brain Repository. Approximately 20 mg of pulverized material was used for ATAC-seq. Frozen samples were thawed in 1 ml of nuclear isolation buffer (20 mM Tris-HCl, 50 mM EDTA, 5 mM spermidine, 0.15 mM spermine, 0.1% mercaptoethanol, 40% glycerol, pH 7.5), inverted for 5 min to mix, and samples were filtered through Miracloth to remove larger pieces of tissue. Samples were centrifuged at $1100 \times g$ for 10 min at 4 °C. The resulting pellet was washed with 50 µl Reduced Swing buffer, centrifuged again, and supernatant was removed. The final crude nuclear pellet was re-suspended in transposition reaction mix and libraries prepared for sequencing as described in Buenrostro et al[23]. All samples were barcoded, and combined into pools. Each pool contained eight randomly selected samples (selection balanced by case–control status and sex). Each pool was sequenced on two lanes of an Illumina 2500 or 4000 sequencer (San Diego, CA, USA) at the Duke Sequencing and Genomic Technologies shared resource.

**ATAC-seq initial processing**. The raw fastq files were processed through cutadapt (version 1.2.0, http://cutadapt.readthedocs.io)[57] to remove adaptors and low-quality reads. cutadapt-filtered reads were aligned to hg19 using bowtie2 (version 2.1.0, http://bowtie-bio.sourceforge.net/bowtie2)[58] using default parameters. In alignment, all reads were treated as single-read sequences, regardless of whether ATAC-seq libraries were sequenced as single end or paired end. The aligned bam files were sorted using samtools (version 0.1.18, https://github.com/samtools)[59], duplicates removed using Picard MarkDuplicates, and then converted to bed format using BedTools (version: v2.17.0, https://broadinstitute.github.io/picard)[60]. ENCODE blacklist regions were removed (i.e., empirically identified genomic regions that produce artifactual high signal in functional genomic experiments, https://sites.google.com/site/anshulkundaje/projects/blacklists). Narrow open chromatin peaks were called from the final bed files using MACS2, with parameter–nomodel–shift -100–ext 200. For visualization, bigwig files were generated using wigToBigWig (version 4) and bedgraph files were output by MACS2. All data have been submitted and made publicly available on Synapse.

**Identification of sample outliers**. We conducted an empirical analysis to identify outliers. An initial analysis identified eight samples that had only had single-end sequencing (unlike the paired end used for all other samples). These samples were excluded.

**Performance of samples**. A total of 314 libraries were sequenced across 86 lanes of either Illumina 2500 or 4000 and generated 53,556,161,474 sequences, which total 7,839,829,094,924 bp of data. After filtering with cutadapt, 51,639,643,049 (96.4% of total) sequences were aligned by bowtie2 and generated 27,809,813,130 (53.9% of reads entering aligner) uniquely aligned reads, and 21,887,340,983 (42.4% of reads entering aligner) multi-aligned reads. Only 3.76% of reads were not aligned. Within the aligned reads, 20,09,897,1743 reads (38.9% of total aligned) were aligned to the mitochondrial genome. On average, MACS2 generated 20,434 ± 12,322 peak calls for each replicate at an FDR <0.01, 28,399 ± 16,917 peak calls at

an FDR <0.05 and 35,607 ± 20,750 peak calls at an FDR <0.10. Non-redundant fraction (NRF) of each replicate is 0.881 ± 0.032, PCR bottleneck coefficient 1 is 0.933 ± 0.030, and PCR bottleneck coefficient 2 is 19.244 ± 8.295.

**Quantification of open chromatin peaks**. Peaks called at an FDR of 1% in each sample were merged, quantified, and normalized using the diffBind R package. Only peaks with overlapping coordinates observed in ≥2 samples were quantified. All reads were extended to 300 bp prior to the quantification process. For replicate samples, we retained the replicate with the highest fraction of total reads overlapping with peaks (and for ties, highest number of peaks detected). Peaks were then merged and quantified again as described without the lower quality replicates and forcing the peak width to be 300 bp using the summit option in the dba.count function of the diffBind R package. Samples were normalized using the trimmed mean of $M$ values method (TMM).

**Performance of replicates**. We prepared nine ATAC-seq replicate samples from an independent brain sample aliquots. We observed that the normalized read counts of replicates were significantly more correlated within pairs of replicates (mean Spearman's correlation = 0.65) than between unrelated samples (mean Spearmans correlation = 0.42) (P value = 0.0001, Supplementary Fig. 9).

**Enrichment of ATAC-seq samples**. We randomly shuffled our merged peak set (118,152 peaks, 300 bp) to random position in the human mapable genome (http://hgdownload.cse.ucsc.edu/goldenPath/hg19/encodeDCC/wgEncodeMapability/wgEncodeCrgMapabilityAlign100mer.bigWig, filtered to only retain mapability score >0.33, excluding ENCODE blacklisted regions (https://sites.google.com/site/anshulkundaje/projects/blacklists), and regions with a peak detected at 10% FDR in any of our 288 samples). This allowed us to quantify the ATAC-seq enrichment of each sample by dividing the number of reads overlapping our merged peak set by the number of reads overlapping the shuffled peak set. This resulted in a median enrichment of 2.9× and similar enrichment for cases and controls (Supplementary Fig. 2b). We also computed an enrichment score for each peak in each sample. To do so, we set the expected number of reads to be the median number of reads overlapping shuffled peaks. This allowed to quantify the number of samples were a single peak was enriched >1× or more than 2× (Supplementary Fig. 2c). This analysis also allowed us to detect enriched peaks highly enriched in many samples (Supplementary Fig. 2d).

**Ancestry estimation and population stratification**. We wished to capture empirical ancestry using Illumina OmniExpressExome SNP data. These were then available as potential covariates for analyses of differential chromatin accessibility and cQTL. We performed principal component analysis (PCA) of LD-pruned SNP array data using EIGENSOFT[61]. From the eigenvalue scree plot, we determined that five PCs were sufficient to control for effects of population stratification in the GWA data. As such, five PCs were included as covariates in the cQTL analysis.

**Evaluation of variables affecting chromatin peaks**. For each sample, we recorded a comprehensive set of 206 metadata features that could conceivably capture some aspect of sample quality. These features were collected from the ATAC-seq processing pipeline, RNA-seq processing of aliquots from the same brain regions[11], and genome-wide SNP genotyping[11]. For example, the metadata included transposase batch, date processed, date submitted, PCR cycles, mean GC percentage of sequenced reads, number of lanes sequenced, mean mapped read length, subject age at death, sex, diagnosis, PMI, antipsychotic use, history of seizures, and RNA quality. We included 10 ancestry-informative PCs from the genome-wide SNP data. We excluded 19 features with high missingness (e.g., time of death, date of death), 24 features that were invariant in all samples (e.g., ATAC-seq library technician, ATAC-seq data processor, brain region, hemisphere), and 40 features with >5% missing values (e.g., hypertension, body mass index, number of weeks without antipsychotics, tobacco use). To prevent potential over-fitting in downstream analysis, we excluded 18 features with >30 levels (e.g., sequencing batch). We then used the R package mice to impute missing values using the classification and regression trees methodology. One feature (RNA-seq expression profiling efficiency) could not be confidently imputed and was excluded. This resulted in a total of 104 metadata variables (65 numeric, 39 categorical) for each of the samples (Supplementary Data 7). Five variables were deconvolution results estimating the proportions of major cell types in the brain from the RNA-seq data (neuron, astrocyte, oligodendrocyte, microglia, endothelial)[11].

**Covariate selection**. In order to detect covariate for our differential chromatin analysis, we performed linear regression of all meta data variables against the first 20 PCs of the TMM normalized peak quantification. In an iterative process, we selected one variable (preferentially a variable directly related to the ATAC-seq experiment, explaining one of the largest proportion of variance and with few parameters), regressed its effect on the peak quantifications and performed a new PCA independent of the selected variable(s). We repeated this procedure until we could not further remove the effect of Bonferroni significant variables. Two metadata variables (sequencing lane and whether the sample was excluded from

the RNA-seq isolation step) remained associated with PC4, PC9, PC13, and PC15 and could not be corrected because of collinearity with the variable "date submitted." We believe that not correcting for these two variables is unlikely to result in false positives as they were among the least different metadata variables between cases and controls (96th and 89th among 100 meta variables tested). In total, we selected the following variables for our differential chromatin analysis: GC (%), date submitted, ChrM aligned (%), NRF, RNA-seq intronic aligned (%), RNA-seq astrocyte (%), RNA-seq oligodendrocyte (%), PC1 genotype, seizures, age of death, PMI, sex, and diagnosis. We note that the variables related to the RNA-seq experiment likely capture unmeasured variables affecting both RNA quality and DNA quality (sample storage duration for example). For example, the percentage of exonic reads in the RNA-seq experiment is highly correlated with several PCs of the ATAC-seq quantification matrix (Supplementary Fig. 10).

**Association between diagnosis and metadata variables.** As shown in Supplementary Fig. 11, we observed that many numerical variables were partially correlated within a large cluster of sequencing-related variables (PCR cycles, number of trimmed reads, number of uniquely aligned reads, etc.). To reduce the dimensionality of the data, we applied PCA, and observed that the first PC largely captured the sequencing-related variables. We also observed that PCs 1, 4, and 9 were significantly associated with diagnosis (Supplementary Fig. 12), indicating that the structure of the metadata was different in several dimensions between cases and controls. To identify the metadata variables associated with case–control status, we evaluated the associations of all metadata variables (104) with case–control status (135 cases, 137 controls). We performed linear regression for numerical variables (65) and $\chi^2$ tests for categorical variables (38, excluding diagnosis) using R.

As shown in Supplementary Fig. 13, we observed that five numerical and four categorical metadata variables were associated at a Bonferroni significance level with case–control status. These variables were: PMI, clinical dementia rating, atypical antipsychotic use, sample storage box, RNA-seq quality, perimortem antipsychotic use, RNA-seq intronic rate, RNA-seq exonic rate, and RNA-seq 28S/16S. The four RNA-seq-related variables were highly correlated (Supplementary Fig. 14), but not correlated with PMI. The variable most strongly associated with case–control status was PMI (Supplementary Figs. 13, 15).

**Association of metadata with number of ATAC-seq peaks.** In the analyses above, we did not observe any significant case–control differences in the number of ATAC-seq peaks detected nor in the estimated cell-type proportions (from RNA-seq). To identify the variables with an effect on ATAC-seq quality, we performed linear regression for all 100 imputed metadata variables (excluding the number of significant ATAC-seq peak calls). We found that 24 variables were associated at a Bonferroni significance level with the number of peaks detected (Supplementary Fig. 16). The numerical variables significantly associated with the number of peaks calls formed highly correlated clusters (Supplementary Fig. 17), indicating that they captured similar technical variability on the number of peak calls.

Although we did not observe any differences between cases and controls in terms of estimated cell-type proportions (from RNA-seq data), we observed that the estimated astrocyte and neuron proportions significantly predicted the number of peak calls. Interestingly, the estimated proportion of neurons and astrocytes were highly correlated with RNA quality (RIN) and slightly less with the mean percentage of GC sequenced. This analysis provided a set of important technical variables affecting the number of peak calls. Nevertheless, this analysis is not sufficient for selecting covariate for the differential chromatin analysis as a variable might not have an effect on the number of peaks detected but an effect on the quantification of the peaks. Hence, we selected covariates for our differential chromatin analysis by looking at the effect of imputed variables on the PCs of the matrix of TMM normalized peak quantification (see above).

**cQTL analysis.** Association analyses for GWA SNPs and open chromatin peaks were evaluated using fastQTL[62], which utilizes a $\beta$ approximation of permutations to determine significance. Imputed genotype probabilities[11] were converted to dosages for input into fastQTL. Peaks were normalized and regressed on SNP dosage in a 5 kb window, controlling for 10 PCs from PCA of peaks and 5 ancestry PCs from PCA of SNP array data. Only the most significant SNP for each peak was retained. To control for testing multiple peaks, we applied FDR correction[63] on the $\beta$-approximated permutation $P$ values.

**Heterozygous individuals display allele bias.** To provide additional characterization of cQTLs, we analyzed ATAC-seq reads from individuals heterozygous for each of the 600 cQTLs that mapped within an ATAC-seq peak. Allele counts from these individuals show a high degree of allele bias that skews from expected 50:50 split (Fig. 5a). This is significantly different (Pearson's $\chi^2$ test, $P$ value <2.2 × $10^{-16}$) from SNPs that were not identified as cQTLs, which show a higher degree of 50% allele counts (Fig. 5b). We also found that the direction of the bias in ATAC-seq reads from heterozygous individuals is largely in the same direction as the effect size of each cQTL (Fig. 5c). In other words, the allele that has more read counts in heterozygous individuals is also the same allele that displays the most accessible chromatin in individuals that are homozygous for that allele. This provides further

evidence that cQTL variants are likely contributing directly to chromatin accessibility.

**cQTL interaction with schizophrenia.** Two hundred and seventy-two individuals (135 schizophrenia cases and 137 controls) were utilized for the cQTL interaction analysis. This analysis was performed similarly to the cQTL analysis described above, with the addition of a term for the main effect of diagnosis and a term where the SNP genotype interacted with schizophrenia diagnosis. Ten thousand permutations were performed to estimate significance, and the Storey and Tibshirani correction was applied to the exact $P$ values[63].

**Overlap of eQTLs with cQTLs.** To determine the true proportion of cQTLs that were also eQTLs, we performed a separate FDR calculation using only the SNPs that were present in both analyses, with the $q$ value package in R. Because there were multiple eQTLs in a gene, we randomly kept one, matched it to the corresponding cQTL, and performed the FDR correction using that subset of cQTL $P$ values. We also repeated this procedure twice keeping only the most (round 1) or least (round 2) significant eQTL per gene to obtain a range of estimates for the true proportion cQTLs that were also eQTLs. Choosing one random eQTL per gene gave an estimate in between these two extreme estimates, as expected. Conversely, we attempted to estimate the true proportion of eQTLs that were also cQTLs. However, all of the nominal eQTL $P$ values corresponding to a significant cQTL were <0.10. Because the distribution was truncated, we could not accurately estimate the true proportion.

**Colocalization analysis.** Analyses were performed starting from the 6200 significant cQTLs that were identified from fastQTL[62]. For colocalization of cQTLs with schizophrenia GWAS, genomic loci significantly associated with schizophrenia were defined as the 108 regions reported in the PGC mega-analysis[5]. For colocalization of cQTLs with schizophrenia eQTLs, analyses were restricted to the regions demonstrating colocalization between cQTLs and GWAS loci. For each of those regions, DLPFC eQTLs[11] that overlapped those cQTL regions of open chromatin were analyzed for colocalization. Bayesian colocalization analysis was performed using the R package coloc[64] to identify regions with evidence for a shared causal variant. When testing two traits, there are five possible hypotheses: no association ($H_0$), association with schizophrenia but not chromatin accessibility ($H_1$), association with chromatin accessibility but not schizophrenia ($H_2$), two distinct causal variants, each associated with one trait ($H_3$), or a shared causal variant that is associated with both schizophrenia and chromatin accessibility ($H_4$). Summary statistics from both the schizophrenia GWAS and cQTL analyses were utilized for analysis and regions with posterior probability ($H_4$) >0.90 were deemed to colocalize.

**Hi-C analysis.** Easy Hi-C (eHi-C) was performed on six postmortem samples ($N$ = 3 adult temporal cortex and $N$ = 3 fetal cerebra, generating over 5 billion total reads (uniquely mapped to hg19, PCR duplicates removed). Following quality control, over 1.323 billion usable reads remained (intra-chromosomal reads mapping >15 kb apart). Human Hi-C data obtainable as of 10/2017 were processed through the same pipeline:[65] 2 adult brain, 12 adult non-brain tissue, and 7 cell line datasets from Schmitt et al.[65], and paired germinal zone and cortical plate samples from 3 fetal brains from Won et al.[12]. After combining replicates, there were 25 Hi-C datasets: new adult and fetal brain, DLPFC, hippocampus, fetal germinal zone, fetal cortical plate, 12 adult tissues (e.g., aorta, spleen, psoas muscle), and 7 cell lines (e.g., IMR90 and GM12878). For the eight regions containing evidence for a shared causal variant between schizophrenia and chromatin accessibility, high confidence eHi-C interactions were new adult and fetal brain, fetal cortical plate, and fetal germinal zone. We visualized looping in the colocalized regions using the Epigenome Browser. To explore the possibility that patterns of chromatin interactions could be different across tissue type, we also visualized the data for each tissue type separately.

**Differential chromatin analysis.** We used DESeq2[66] to detect differential chromatin. The primary analysis was for case–control status, but we also evaluated, age at death, and PMI. The following variables were included in the model: GC (%), date submitted, ChrM aligned (%), NRF, RNA-seq Intronic aligned (%), RNA-seq astrocyte (%), RNA-seq oligodendrocyte (%), PC1 genotype, seizures, age at death, PMI, sex, and diagnosis. All 288 samples were used for this analysis as this increases power to correctly estimate the parameters of the model. The case–control difference was performed between control individuals ($N$ = 135) and individuals with schizophrenia ($N$ = 137). Although we corrected for sex in all our analysis, we observed that peaks located on the sex chromosomes were more often called significant than peaks located on autosomes. We performed sex-stratified analysis for case/control difference in open chromatin and did not observe that peaks on sex chromosomes were enriched in low $P$ values compared to other chromosomes. Therefore, we believe that the initial enrichment in significant hits for all analysis (case/control, sex, PMI, and age) on the X and Y chromosomes were likely false positives. In order to prevent potential bias due to the sex chromosomes, we meta-analyzed our sex-stratified differential chromatin analysis on chrX using an inverse variance-weighted approach[67] and only used $P$ values obtained in male

for chrY. For the remaining chromosomes, the full model was used. This approach resulted in the detection of three peaks differentially accessible between cases and controls (5% FDR) that we report in the main text.

We also assessed other potential strategies for differential chromatin analysis in order to assess the robustness of our result. First, we selected 60 cases and 60 controls with PMI <18 h and matched for PMI, age at death, sex, and mean GC content. Since the 60 cases and 60 controls were matched for the most important variables affecting peak quantifications, we used DESeq2 without correcting for any covariates. Under this model, we did not observe any significant difference between cases and controls at an FDR of 5% (top hit had a $q$ value of 0.81). Second, we selected 41 cases and 41 controls with PMI <24 h, with the total number of peaks detected greater than the median that were matched for PMI, age of death, GC content, sex and the number of peaks detected at 1% FDR. Again, we did not find any significant differences (5% FDR) between cases and controls using this strategy (top hit had a $q$ value of 0.25). Third, we performed a new peak merging procedure with stricter criteria. We required that the fraction of reads mapping to peaks within each sample was at least 1% and that at least 10K peaks were discovered within each sample. These criteria excluded 38 samples of lower quality. We then merged peaks of the 250 remaining samples without any width restriction and only kept peaks overlapping in at least 25 samples. This procedure lead to the quantification of 41,731 high confidence peaks. We matched cases/controls on the most important variables (PMI, age of death, sex, GC content, and number of peaks detected) from 110 schizophrenia cases and 110 matched controls. Again, we did not observe any differences between cases and control using this stricter threshold for peak calling (top hit with a $q$ value of 0.46). Finally, we used the resonance ultrasonic vibration (RUV)[68,69] methodology to capture unmeasured confounders. Briefly, we ran a differential chromatin analysis testing for an effect of case/control status (DESeq2) using UIC replication samples (see below for description) that had an average enrichment in our peaks calls >2-fold compared to randomly shuffled peaks (total samples: 15 schizophrenia cases, 170 controls). We used the following covariates for the differential chromatin analysis: BrainBank + PMI + Sex + Age at death + RIN + enrichment score, and detected no significant results (top $q$ value = 0.47). We then took the 20,000 peaks with least significant $P$ values (testing for an effect of diagnosis) as our set of external true negatives for RUV. We then learned 10 factors with RUV on the selected peaks (in our dataset) and performed the differential chromatin analysis correcting for these 10 factors. This lead to the discovery of six differentially accessible peaks at 5% FDR (all three of the original differential peaks were included in the six discovered using RUV).

**Gene enrichment analysis.** We used GREAT[70] to test for biological enrichment of genes located in close proximity to our differentially accessible peaks. We used all tested peaks as background.

**Gene-set enrichment analysis.** We used MAGMA[40] to test whether the closest genes to our differentially accessible peaks were enriched in GWAS association statistics with schizophrenia. MAGMA allows to combine summary statistics of SNPs into a gene-level $P$ values (in an LD aware manner). In a second step, a linear regression is performed to test whether the gene set is more significantly associated with the trait than the rest of the genome (competitive test). In order to compute gene-level association statistics, we set a window of 10 kb upstream to 1.5 kb downstream of each gene and used the summary statistics from the PGC2 GWAS of schizophrenia[5].

**Heritability enrichment analysis.** We used LD score regression to estimate heritability enrichment in our ATAC-seq peak calls[25]. We added SNPs located in our ATAC-seq peaks to the baseline model which consists of 53 categories representing different genomic annotations (TSS, promoter, enhancer, CTCF binding sites, etc.). In addition, we added SNPs located within and around our ATAC-seq peaks (500 bp upstream and 500 downstream) in an extra annotation to prevent upward bias in the heritability enrichment[25]. We note that this step is recommended by the authors of LD score regression and is only used to accurately estimate the heritability of the region of interest (here our 300 bp peaks).

In a second analysis, we added SNPs that were present both in the ATAC-seq peak calls and in the conserved regions as an extra annotation to the model described above (baseline model + ATAC-seq peaks + ATAC-seq peak ± 500 bp extended annotation). This allowed us to estimate the heritability enrichment of regions that are conserved across mammals and present in brain open chromatin regions.

In a third analysis, we tested the effect of increasing peak width on the heritability enrichment of the peaks. We tested 100 bp, 300 bp, 1 kb, 2 kb, 5 kb, and 10 kb peaks. For each peak size, we used the 53 annotations from the baseline model, the ATAC-seq peaks (of the specific size of interest) and an annotation surrounding the ATAC-seq peak of interest by 500 bp on both sides.

To compare the association of schizophrenia across different tissues, we added SNPs falling into the respective open chromatin region to the baseline model (without the 500 bp windows) and used the coefficient $z$-score as a measure of association between the annotation and schizophrenia, as recommended[25].

**Amount of coverage between DNase-seq/ATAC-seq data.** For each DNase-seq and ATAC-seq dataset from 125 tissues, we calculated the total number of bases covered (Supplementary Fig. 18) and the average Jaccard index (an indicator of overall similarity of datasets, Supplementary Fig. 19). Jaccard index indicates that ATAC-seq data is most similar to ATAC-seq data from sorted neuronal (NeuN+) and glial (NeuN−) cells (Supplementary Fig. 20).

**Motif enrichment.** We intersected conserved regions[26] with our ATAC-seq peaks. As MEME-chip requires all input sequences to have the same length, we set the width of each intersected region to 32 bp (16 bp upstream and downstream of the center of the intersected region, corresponding to the mean size of the intersected regions). We obtained hg19 sequence corresponding to these regions using bedtools. We then used MEME-chip[60] to look for motif enrichment using 32 bp around the center of all ATAC-seq peaks as background, and used HOMER[71] (v4.9) with intersected conserved regions with our ATAC-seq peaks as input and all ATAC-seq peaks as background.

**Comparison with ATAC-seq from sorted nuclei.** We compared our peaks with ATAC-seq peaks from sorted neuronal (NEU+) and non-neuronal (NEU−) nuclei from the prefrontal cortex (Brodmann area 10)[22]. We found a larger jaccard index (intersection/union of bed files) between our ATAC-seq peaks and NEU+ peaks (0.14) than between our peaks and the NEU− peaks (0.1), suggesting that the proportion of neuron-derived peaks is higher than glial-derived peaks in our samples (Supplementary Fig. 20). Using LD score regression, we did not find significant heritability enrichment for schizophrenia in peaks from sorted nuclei (NEU+, heritability enrichment = 3.1×, $P$ value = 0.1) (NEU−, heritability enrichment = 1.7×, $P$ value = 0.54) or the union of NEU+ and NEU− peaks (heritability enrichment = 1.9×, $P$ value = 0.33). Of our 118,152 merged peaks, we found that 33,242 overlapped peaks in the NEU− samples, 42,599 overlapped peaks in the NEU+ fraction and 58,377 peaks overlapped peaks in either the NEU + or NEU− ATAC-seq peaks. Finally, we found that 59,775 peaks were unique to our study and these were highly enriched for schizophrenia heritability (heritability enrichment = 9.6×, $P$ value = 0.016).

**Replication dataset.** The Chicago dataset consists of dorsolateral prefrontal cortexes from 47 individuals with schizophrenia and 218 controls. We directly obtained the sequencing files (.fastq) from the University of Chicago. We then mapped reads and called peaks using the same pipeline as for our samples (merged peaks detected at a 1% FDR, only kept peaks observed in at least two samples and set peak width to 300 bp). We detected 157,660 peaks using the Chicago samples. We found that 86% of our peaks overlapped with the Chicago merged peak set (65% of the peaks detected using their samples overlapped with ours). These 157,660 peaks were used to replicate the schizophrenia heritability enrichment that we observed using our peaks. For the replication of our differential chromatin analysis results, we quantified the ATAC-seq enrichment of the Chicago samples in our merged peak set (see methods: Enrichment of ATAC-seq samples methods) and used samples with and enrichment score >2× (15 cases/170 controls). The differential chromatin analysis was then performed as described above using the following covariates: BrainBank + PMI + Sex + Age at death + RIN + enrichment score.

**Genome build.** All genome coordinates are GRCh37/hg19.

**Code availability.** Computer code is available upon request.

**Data availability.** These ATAC-seq data are available from Sage Bionetworks-Synapse website via the psychENCODE Knowledge Portal under the accession number [syn5321694] https://www.synapse.org/#!Synapse:syn5321694. A table of the processed samples is also available at: https://www.synapse.org/#!Synapse:syn12214341/tables/.

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

## Acknowledgements

This project was funded by the US NIMH (R01 MH105472, PI Crawford, co-PI Sullivan) and NIGMS (P30GM103328, PI Stockmeier). P.F.S. was supported by the Swedish Research Council (Vetenskapsrådet, award D0886501). J.B. was supported by the Swiss National Science Foundation. We are deeply indebted to Vahram Haroutunian, Pamela Sklar, and the CommonMind Consortium for providing brain samples. We also thank the NICHD tissue bank (Baltimore, MD, USA) for additional postmortem human tissues, and the NIH NeuroBioBank for fetal brain samples. This work is part of the psychENCODE Consortium, which was funded by US NIH grants: U01MH103339, U01MH103365, U01MH103392, U01MH103340, U01MH103346, R01 MH105472, R01MH094714, R01MH105898, R21MH102791, R21MH105881, R21MH103877, and P50MH106934 awarded to: Schahram Akbarian (Icahn School of Medicine at Mount Sinai), Gregory Crawford (Duke), Stella Dracheva (Icahn School of Medicine at Mount Sinai), Peggy Farnham (USC), Mark Gerstein (Yale), Daniel Geschwind (UCLA), Thomas M. Hyde (LIBD), Andrew Jaffe (LIBD), James A. Knowles (USC), Chunyu Liu (UIC), Dalila Pinto (Icahn School of Medicine at Mount Sinai), Nenad Sestan (Yale), Pamela Sklar (Icahn School of Medicine at Mount Sinai), Matthew State (UCSF), Patrick Sullivan (UNC), Flora Vaccarino (Yale), Sherman Weissman (Yale), Kevin White (U Chicago), and Peter Zandi (JHU). Data were also generated as part of the CommonMind Consortium and supported by funding from Takeda Pharmaceuticals Company Limited, F. Hoffman-La Roche Ltd, and NIH grants R01MH085542, R01MH093725, P50MH066392, P50MH080405, R01MH097276, RO1-MH-075916, P50M096891, P50MH084053S1, R37MH057881 and R37MH057881S1, HHSN271201300031C, AG02219, AG05138, and MH06692. Brain tissue for the study was obtained from the following BrainBank collections: the Mount Sinai NIH Brain and Tissue Repository, the University of Pennsylvania Alzheimer's Disease Core Center, the University of Pittsburgh NeuroBioBank and Brain and Tissue Repositories, and the NIMH Human Brain Collection Core. CMC Leadership: Pamela Sklar, Joseph Buxbaum (Icahn School of Medicine at Mount Sinai), Bernie Devlin, David Lewis (University of Pittsburgh), Raquel Gur, Chang-Gyu Hahn (University of Pennsylvania), Keisuke Hirai, Hiroyoshi Toyoshiba (Takeda Pharmaceuticals Company Limited), Enrico Domenici, Laurent Essioux (F. Hoffman-La Roche Ltd), Lara Mangravite, Mette Peters (Sage Bionetworks), Thomas Lehner, Barbara Lipska (NIMH). We also thank the Duke Sequencing and Genomic Technologies Core facility for sequencing the ATAC-seq libraries. We thank Mette Peters for her work with uploading and making our data available on Sage Bionetworks.

## Author contributions

J.B., M.E.G., L.S., A.A.-K., P.F.S., and G.E.C. designed the study and wrote the manuscript. J.B. performed quality control, peak quantification, covariate selection, heritability enrichment analysis, motif enrichment analysis with MEME-chip, and differential chromatin analysis. M.E.G. performed the sample identity analysis, cQTL analysis, and colocalization analysis. L.S. processed raw data, performed peak calling, motif enrichment analysis using HOMER, and allele-specific analysis for cQTLs. A.S. performed the ATAC-seq experiment. A.S., P.G.-R., G.D.J., A.B., J.F.F., P.R., G.A.W., and T.E.R. generated raw data and interpreted findings. P.F.S., S.A., V.H., C.A.S. acquired brain samples and interpreted metadata. K.P.W. and C.L. generated the UIC ATAC-seq dataset. All authors read and approved the manuscript.

## Additional information

**Competing interests**: P.F.S. is a scientific advisor for Pfizer, Inc. G.E.C. and T.E.R. are co-founders of Element Genomics, Inc. The remaining authors declare no competing interests.

