## [Peer Review File · Nature Communications]

Reviewers' comments:

Reviewer #2 (Remarks to the Author):

The authors have addressed thoroughly all of my questions.

Reviewer #3 (Remarks to the Author):

Bryois et al sufficiently addressed my comments.

However, in responding to Reviewer 1, I do not think the current implementation of RUV is correct. In the methods, the authors describe selecting control features based on lack of association with the primary outcomes of interest (e.g. diagnosis): "We obtained 5 factors of unwanted variability ($k=5$) on the 50'000 least associated peaks between cases and controls in our primary differential expression analysis (min pvalue= 0.58)". This selection of "negative control features" unfortunately biases the differential binding analysis, as true negative control features should be selected a priori, as described in the original RUV paper (Gagnon-Bartsch & Speed, Biostatistics 2012).

In fact, those authors warn against this exact approach in their Discussion: "There may be a temptation to "discover" negative control genes. For example, a researcher may wish to find genes whose expression levels are not highly correlated with the factor of interest, label these genes as negative controls, and then adjust via RUV-2. The allure of this approach is clear—finding a set of negative controls would be much easier and could in fact be automated. However, we feel this approach is misguided. If there are unwanted factors that are correlated with the factor of interest, then the expression levels of the true negative controls should in fact be correlated with the factor of interest. Excluding genes correlated with the factor of interest would bias our estimate of the unwanted factors."

If the authors want to implement RUV, I think it would be necessary to determine the peaks/regions of the genome most associated with technical factors in an independent dataset. And then the authors would perform factor analysis on the coverage of these regions in their dataset (regardless of if they are actually defined as peaks). This is analogous to an approach described for modeling RNA degradation in postmortem brain studies [PMID: 28634288]. Otherwise, the authors should remove the current RUV analyses (as they are biased) and retain their original differential binding analyses.

Reviewer #4 (Remarks to the Author):

The paper by Bryois, Garrett, and Song et al. describes the analysis of ATAC-seq data from dorso-lateral prefrontal cortex from a large cohort of 135 cases with schizophrenia and 137 controls. The authors use this data to explore the mechanisms of non-coding regulatory elements in schizophrenia with a focus on better understanding published GWAS data. This work represents one of, largest efforts to characterize the open chromatin landscapes of a psychiatric disease. The authors provide detailed methods which are important to understand the relatively complex set of analyses performed. While I view this work represents a substantial investigation on an important topic, a number of concerns preclude my recommendation of this article for publication in Nature Communications.

Major concerns:

1) The overall data quality appears to be quite low. The authors state that approximately 2-6% of uniquely aligned reads fall within 300 bp peaks. This number is lower than the standard in the

field. I'm sure that this likely results from the type of tissue used (pulverized frozen tissue) and there is nothing to be done about the data quality at this point. However, I believe the paper would benefit from more sequencing track figures similar to those shown in Fig. 1B. For example, tracks at additional important SNPs that are mentioned throughout the paper and examples of peaks/regions that are differentially accessible between schizophrenia and controls.

2) Along the same lines, I share some of the concerns about FRiP expressed by Reviewer #2 during the first round of review. In the current rebuttal, the authors state that the FRiP obtained in the brain samples from this study is similar to the FRiP obtained from other tissues. It would perhaps be more informative to see how the FRiP compares to other published ATAC-seq datasets from brain tissue (PMID: 28335009 [by coauthors on this manuscript] and/or PMID: 28846090).

3) Overall, the biological insights gained from this large cohort study are unclear. The authors should endeavor to provide more concrete findings. In its current form, the bulk of the paper reads as a list of statistics which could benefit from an explanation of biological relevance. For example, much of the motif enrichment analysis is reported as a list of motifs without attaching a potential biological explanation for the observation.

4) One of the novel aspects of this cohort is the associated RNA and genotype information available. It would be interesting to see (for example) how the schizophrenia-specific peaks affect nearby gene expression and whether the expression of these nearby genes shows the same correlation across schizophrenia and controls.

Minor concerns:

5) There are statements throughout the paper that do not contain figure/table citations. For example "This distribution was similar to previous studies using DNase-seq" on Page 3. It is impossible for the reader to assess the meaning of "similar" without seeing the distributions. This is just one example but the authors should check the manuscript for similar situations as I encountered multiple.

6) The authors state that the SNP-heritability enrichment of DLPFC ATAC-seq peaks was specific to schizophrenia and was not significantly enriched for GWAS variants for educational attainment, height, or total cholesterol. What about other brain-related GWAS such as alzheimers, autism, ALS, bipolar disorder, depression, etc.

7) The authors state that conserved regions in ATAC-seq peaks were significantly enriched for CTCF binding and go on to discuss the potential importance topological associated domains (in both the results and discussion sections). ATAC-seq peaks in general are enriched for CTCF binding sites so it would be important to know if this enrichment is greater than a background set of ATAC-seq peaks. If this is actually the statistic being reported, the authors should make that more clear.

We greatly appreciate the reviewers' comments and additional suggestions. Our responses to their questions are below in **blue text**. We also have provided a revised manuscript with changes in **blue text**.

Reviewers' comments:

Reviewer #2 (Remarks to the Author):

The authors have addressed thoroughly all of my questions.

We appreciate that we've addressed the reviewer's comments.

Reviewer #3 (Remarks to the Author):

Bryois et al sufficiently addressed my comments.

We appreciate that we've addressed the reviewer's comments.

However, in responding to Reviewer 1, I do not think the current implementation of RUV is correct. In the methods, the authors describe selecting control features based on lack of association with the primary outcomes of interest (e.g. diagnosis): "We obtained 5 factors of unwanted variability (k=5) on the 50'000 least associated peaks between cases and controls in our primary differential expression analysis (min pvalue= 0.58)". This selection of "negative control features" unfortunately biases the differential binding analysis, as true negative control features should be selected a priori, as described in the original RUV paper (Gagnon-Bartsch & Speed, Biostatistics 2012).

In fact, those authors warn against this exact approach in their Discussion: "There may be a temptation to "discover" negative control genes. For example, a researcher may wish to find genes whose expression levels are not highly correlated with the factor of interest, label these genes as negative controls, and then adjust via RUV-2. The allure of this approach is clear—finding a set of negative controls would be much easier and could in fact be automated. However, we feel this approach is misguided. If there are unwanted factors that are correlated with the factor of interest, then the expression levels of the true negative controls should in fact be correlated with the factor of interest. Excluding genes correlated with the factor of interest would bias our estimate of the unwanted factors."

If the authors want to implement RUV, I think it would be necessary to determine the peaks/regions of the genome most associated with technical factors in an independent dataset. And then the authors would perform factor analysis on the coverage of these regions in their dataset (regardless of if they are actually defined as peaks). This is analogous to an approach

described for modeling RNA degradation in postmortem brain studies [PMID: 28634288]. Otherwise, the authors should remove the current RUV analyses (as they are biased) and retain their original differential binding analyses.

We thank Reviewer 3 for their point about the original RUV paper and agree that the RUV analysis that we performed may be biased. As a follow up, we ran an unbiased RUV differential chromatin analysis and found very similar results to our original analysis. To perform the unbiased differential chromatin results using RUV, we ran a differential chromatin analysis (DESeq2) using the Chicago schizophrenia and control brain samples (external dataset) that had an average enrichment in our peaks calls >2 fold compared to randomly shuffled peaks (total samples: 15 schizophrenia cases, 170 controls). We used the following covariates for the differential chromatin analysis: BrainBank + PMI + Sex + Age at death + RIN + enrichment score, and detected no significant associations in the external dataset (top qvalue = 0.47). We then took the 20,000 peaks with least significant p-values (testing for an effect of diagnosis) as our set of external true negatives for RUV. We then learned 10 factors with RUV using the selected peaks (in our dataset) and performed the differential chromatin analysis correcting for these 10 factors. This led to the discovery of 6 differentially accessible peaks at 5% FDR versus 3 differentially accessible peaks in our original analysis (all 3 of the original differential peaks were included in the 6 discovered using the unbiased RUV analysis). In light of these findings, we have decided to revert to our original differential chromatin analysis as this allows us to use the Chicago dataset as a replication dataset.

In addition to the above analysis, we also applied multiple alternative strategies for our differential chromatin analysis, that support there being little evidence for differences between cases and controls. First, we selected 60 cases and 60 controls with PMI < 18h and matched for PMI, age at death, sex and mean GC content. Since the 60 cases and 60 controls were matched for the most important variables affecting peak quantifications, we used DESeq2 without correcting for any covariates. Under this model, we did not observe any significant difference between cases and controls at an FDR of 5% (top hit had a qvalue of 0.81). Second, we selected 41 cases and 41 controls with PMI < 24h, with the total number of peaks detected greater than the median that were matched for PMI, age of death, GC content, sex and the number of peaks detected at 1% FDR. Again, we did not find any significant differences (5% FDR) between cases and controls using this strategy (top hit had a qvalue of 0.25). Third, we performed a new peak merging procedure with stricter criteria. We required that the fraction of reads mapping to peaks within each sample was at least 1% and that at least 10K peaks were discovered within each sample. These criteria excluded 38 samples of lower quality. We then merged peaks of the 250 remaining samples without any width restriction and only kept peaks overlapping in at least 25 samples. This procedure led to the quantification of 41,731 high confidence peaks. We matched cases/controls on the most important variables (PMI, age of death, sex, GC content and number of peaks detected) from 110 schizophrenia cases and 110 matched controls. Again, we did not observe any difference

between cases and control using this stricter threshold for peak calling (top hit with a qvalue of 0.46).

Altogether, our results indicate that there are little differences in chromatin accessibility between cases and controls. This is in contrast to there being a substantial number of differential peaks in adult post-mortem DLPFC brain samples that are due to the effect of age (2310 peaks at 5% FDR) and post-mortem interval (466 peaks at 5% FDR). We believe this is an interesting finding, and points to other possible mechanisms that can be studied in subsequent manuscripts.

Reviewer #4 (Remarks to the Author):

The paper by Bryois, Garrett, and Song et al. describes the analysis of ATAC-seq data from dorso-lateral prefrontal cortex from a large cohort of 135 cases with schizophrenia and 137 controls. The authors use this data to explore the mechanisms of non-coding regulatory elements in schizophrenia with a focus on better understanding published GWAS data. This work represents one of, largest efforts to characterize the open chromatin landscapes of a psychiatric disease. The authors provide detailed methods which are important to understand the relatively complex set of analyses performed. While I view this work represents a substantial investigation on an important topic, a number of concerns preclude my recommendation of this article for publication in Nature Communications.

Major concerns:

1) The overall data quality appears to be quite low. The authors state that approximately 2-6% of uniquely aligned reads fall within 300 bp peaks. This number is lower than the standard in the field. I'm sure that this likely results from the type of tissue used (pulverized frozen tissue) and there is nothing to be done about the data quality at this point. However, I believe the paper would benefit from more sequencing track figures similar to those shown in Fig. 1B. For example, tracks at additional important SNPs that are mentioned throughout the paper and examples of peaks/regions that are differentially accessible between schizophrenia and controls.

As requested, we have included a new Figure S6 that shows a box plot of the three differentially accessible peaks (5% FDR) between cases and controls. Since these are subtle changes, we believe a box plot most accurately represents the data.

2) Along the same lines, I share some of the concerns about FRiP expressed by Reviewer #2 during the first round of review. In the current rebuttal, the authors state that the FRiP obtained in the brain samples from this study is similar to the FRiP obtained from other tissues. It would perhaps be more informative to see how the FRiP compares to other published ATAC-seq datasets from brain tissue (PMID: 28335009 [by coauthors on this manuscript] and/or PMID: 28846090).

We thank the reviewer for this comment, and have now included both FRiP scores for our tissue homogenate data, as well as from sorted NeuN+/- data from PMID 28335009 as a new Figure S3B. Compared to our data, we find that FRiP scores from PMID 28335009 are about 2x higher. We believe there are a number of factors that may contribute to the difference in FRiP scores. First, we know that tissue homogenate is a combination of cell types, and our ATAC-seq signal is therefore likely distributed among cell types. This undoubtedly reduces the overall FRiP scores since the signal is distributed among a larger set of weaker peaks. Second, nuclei sorting enriches for intact nuclei, while using tissue homogenate characterizes all material, including intact nuclei, partially intact nuclei, and potentially naked DNA fragments.

Together, it is not surprising that PMID 28335009 using sorted nuclei has higher FRiP scores. However, we note that peaks that were discovered only in our tissue homogenate data are highly enriched for SCZ heritability, while the sorted NeuN+/- ATAC-seq data collected for PMID 28335009 did not show the same enrichment, even when we characterize the union set of all NeuN+/- ATAC-seq peaks. We have included the following paragraph in the supplement methods section.

Comparison with ATAC-seq from sorted nuclei. We compared our peaks with ATAC-seq peaks from sorted neuronal (NEU+) and non-neuronal (NEU-) nuclei from the prefrontal cortex (Brodmann area 10)¹²⁹. We found a larger jaccard index (intersection/union of bed files) between our ATAC-seq peaks and NEU+ peaks (0.14) than between our peaks and the NEU- peaks (0.1), suggesting that the proportion of neuron-derived peaks is higher than glia-derived peaks in our samples (Figure S20). Using LDscore regression, we did not find significant heritability enrichment for schizophrenia in peaks from sorted nuclei (NEU+, heritability enrichment = 3.1x, pvalue=0.1) (NEU-, heritability enrichment = 1.7x, pvalue=0.54) or the union of NEU+ and NEU- peaks (heritability enrichment = 1.9x, pvalue=0.33). Of our 118,152 merged peaks, we found that 33,242 overlapped peaks in the NEU- samples, 42,599 overlapped peaks in the NEU+ fraction and 58,377 peaks overlapped peaks in either the NEU+ or NEU- ATAC-seq peaks. Finally, we found that 59,775 peaks were unique to our study and these were highly enriched for schizophrenia heritability (heritability enrichment = 9.6x, pvalue=0.016).

This indicates that our data from characterizing intact homogenate tissue is capturing important information that is lost during the nuclei sorting procedure. In other words, it is unknown if the nuclei sorting procedure is biased for only enriching for the hardiest of nuclei that are able to maintain integrity during the sorting process. It is possible that certain types of nuclei from certain types of cells in the brain are depleted during this procedure, or chromatin structure somehow changes during the sorting process.. This is an important finding, and will be the focus of follow up studies.

3) Overall, the biological insights gained from this large cohort study are unclear. The authors should endeavor to provide more concrete findings. In its current form, the bulk of the paper reads as a list of statistics which could benefit from an explanation of biological relevance. For

example, much of the motif enrichment analysis is reported as a list of motifs without attaching a potential biological explanation for the observation.

We appreciate this comment, and have added additional text regarding motifs that that are enriched in our analysis. We agree these are likely candidates for follow up studies.

4) One of the novel aspects of this cohort is the associated RNA and genotype information available. It would be interesting to see (for example) how the schizophrenia-specific peaks affect nearby gene expression and whether the expression of these nearby genes shows the same correlation across schizophrenia and controls.

We quantified the chromatin accessibility of each promoter (2kb upstream to TSS) of each gene in GENCODE v25 (53,224 genes). We performed a differential chromatin analysis for each promoter using the same covariates as our original differential chromatin analysis and did not observe any significant differences between cases and controls (top qvalue=0.297). Restricting to protein coding genes for multiple testing comparison did not lead to any significant results (top qvalue=0.72). Restricting to genes found to be differentially expressed between cases and controls in the same samples by the CommonMind consortium (693 at 5% FDR), also did not result in any differential accessibility (top qvalue=0.7). However, we show that genes with open chromatin around the TSS tend to show higher level of expression (Figure 1C). Given that the 693 genes were not enriched in any biological functions and nor are they enriched for genetic association with schizophrenia (using MAGMA and SCZ GWAS), it remains unclear whether this list of genes are true positives and whether we should expect to see a difference in chromatin accessibility around these genes.

4613 ATAC-seq peaks were located within CommonMind consortium differentially expressed genes. Correcting for multiple testing only within these genes did not lead to any significant findings (top qvalue=0.77)). Finally, we found that, as a group, the 4613 peaks located within differentially expressed genes did not have lower pvalues for case/control comparison than the rest of the peaks (p=0.5).

Altogether, we show that genes with higher level of expression tend to have more open chromatin around the TSS (Figure 1), however we find no evidence that chromatin accessibility differs between cases and controls for genes that were called to be differentially expressed by the CommonMind Consortium. This may indicate that certain co-factors that regulate gene expression are differentially present between cases and controls (and not detectable by ATAC-seq), that there may be other post-transcriptional regulation processes at work (e.g., RNA stability), or there may be some other unknown mechanism. Regardless, this study is an important first step in characterizing these samples.

Minor concerns:

5) There are statements throughout the paper that do not contain figure/table citations. For example “This distribution was similar to previous studies using DNase-seq” on Page 3. It is impossible for the reader to assess the meaning of “similar” without seeing the distributions. This is just one example but the authors should check the manuscript for similar situations as I encountered multiple.

We thank the reviewer for his comment. We have now clarified two sentences that we found unclear (added text below underlined).

1. **“This distribution was similar to previous studies using DNase-seq (13% at TSS \pm 2 kb, 26% within the gene body and 34% intergenic).”**
2. **“ The estimated proportion of significant cQTLs that are also eQTLs is 23.3% (Online Methods), which agrees with similar estimates from lymphoblastoid cell lines (23%). ”**

6) The authors state that the SNP-heritability enrichment of DLPFC ATAC-seq peaks was specific to schizophrenia and was not significantly enriched for GWAS variants for educational attainment, height, or total cholesterol. What about other brain-related GWAS such as alzheimers, autism, ALS, bipolar disorder, depression, etc.

We thank the reviewer for this suggestion. The authors of LDSC recommend at least 5000 cases before attempting to estimate heritability and much more (not specified) for partitioning heritability. In addition, the statistical power for heritability enrichment depends on the size of the tested genomic annotation (illustrated in our Fig. 2C). Since our ATAC-seq peaks represent ~1% of the genome, which is small compared to other genomic annotations of the “baseline model” of LDSC, we decided to estimate heritability enrichment in our peaks only with the most well powered GWAS available.

We do not believe that the currently published GWAS for Alzheimer, autism, ALS, bipolar disorder and depression are sufficiently powered for heritability enrichment in small genomic regions. However, we decided to add to our manuscript a recent well powered GWAS of cognitive ability (<https://www.biorxiv.org/content/early/2017/09/06/184853.1> , in press in Nature Genetics).

7) The authors state that conserved regions in ATAC-seq peaks were significantly enriched for CTCF binding and go on to discuss the potential importance topological associated domains (in both the results and discussion sections). ATAC-seq peaks in general are enriched for CTCF binding sites so it would be important to know if this enrichment is greater than a background set of ATAC-seq peaks. If this is actually the statistic being reported, the authors should make that more clear.

We thank the reviewer for pointing out that we were not sufficiently clear in the text. We used two different motif enrichment algorithm (MEME and HOMER). For MEME, we used the default setting, where MEME constructs a background based on the submitted sequence, while for HOMER we used as background 100,000 random conserved regions. We realized that our background set may not have been optimal and decided to update

our analysis and use all our ATAC-seq peaks for motif enrichment analysis for both MEME and HOMER.

REVIEWERS' COMMENTS:

Reviewer #3 (Remarks to the Author):

The authors have addressed all of my comments

Reviewer #4 (Remarks to the Author):

The authors have adequately addressed my concerns.